# Wetland Change Mapping Using Machine Learning Algorithms, and Their Link with Climate Variation and Economic Growth: A Case Study of Guangling County, China

Gadisa Fayera Gemechu [1,2] , Xiaoping Rui [3,*] and Haiyue Lu [1,*]

1   College of Hydrology and Water Resources, Hohai University, Nanjing 210098, China; gadisa@hhu.edu.cn
2   Faculty of Natural Sciences, Salale University, Fiche 245, Ethiopia
3   College of Earth and Engineering, Hohai University, Nanjing 211100, China
*   Correspondence: ruixp@hhu.edu.cn (X.R.); 191309010002@hhu.edu.cn (H.L.)

**Abstract:** Wetlands are a distinctive terrestrial ecosystem that benefits living things, including people, in various ways. Sustainable wetland ecosystem resources are needed to protect the global environment. Wetlands in China have undergone positive and negative changes in response to several factors, but studies documenting their long-term dynamicity have been few, particularly in Guangling County. This study examines the change of wetlands area based on remotely sensed data while exploring trends associated with climate variations and economic growth in Guangling County, China. Analysis of remotely sensed imagery, mainly in hilly and nonhomogeneous environments is problematic, largely as a result of interference and their high spectral non-homogeneity. We conducted experiments using five classical machine learning algorithms based on the Google Earth Engine (GEE) and obtained the greatest robustness and accuracy using a Support Vector Machine (SVM)—Radial Basis Function (RBF) kernel approach, with overall accuracy and kappa statistics ranging from 86% to 98.1% and from 0.789 to 0.960, respectively. Based on the SVM-RBF model's outperformance of four other algorithms, we identified spatial distributions of wetland in the study area and associated change trends. We found that 45.71 km$^2$ of wetland area was lost over the past 3.7 decades (January 1984–December 2020), or 81.82% of wetland area coverage. In this paper, we explore how factors such as county economic growth (GDP), humidity, and temperature variations are tightly linked with wetland change.

**Keywords:** wetland; change detection; climate; gross domestic product; Google Earth Engine; machine learning; remote sensing; Guangling County

## 1. Introduction

Sustainable wetland ecosystems can offer direct monetary value to human beings and indirectly serve human beings [1]. Wetlands are highly productive natural resources that provide globally substantial social, economic, and environmental benefits. They are regarded as the "kidneys of the Earth" and are its most important ecosystems [2]. Wetlands perform a wide range of stabilizing functions, including by improving water quality; recharging groundwater; storing water and natural products; offering aesthetic and recreational opportunities; mitigating storms and flooding; controlling erosion and stabilizing shorelines; helping support fish, timber, peat, and wildlife resources; and increasing tourism opportunities [3,4]. The key role of many wetlands in supporting biological communities means that they are ecosystems that provide valuable goods. For example, wetlands can provide food, water, and shelter to plants, animals, and humans [5]. As an intact land cover, they are equally associated with environmental monitoring and national economies.

Wetlands are naturally dynamic systems that can be created, re-formed, degraded, and destroyed by a range of natural processes [3]. Natural or artificial wetlands can be lost or formed, and sudden reductions in the degradation of wetlands have become a global

phenomenon in response to population growth and climate change [6]. Wetland loss and degradation are caused mainly by human pressure, which increases demand for agricultural land associated and land reclamation with population growth [4,7,8]. The consequences are very serious: Wetland loss means loss of habitats, decrease in water depth, deterioration in water quality, and damage to natural ecosystems [9]. Since 1950, China's wetlands have faced serious issues that have led to calls for research and monitoring, being under constant threat from multiple influencing factors [10]. During this time, they have undergone a critical change.

Wetland change monitoring is particularly important with a spatial arrangement to understand their ecosystem functions and services, as well as to establish management policies and implementations [11]. Efficiently and accurately assessing change trends of wetlands is particularly crucial for decision making and to mitigate wetland loss [12]. Mapping wetland status is, thus, key to monitoring and development planning before adopting new policies for managing wetland resources [13]. The use of remote sensing technology for change detection based on image differencing (including spectral mixture modeling and various spectral indices) could be a particularly effective way of mapping land cover changes [3]. Analyzing the time series of remote sensing imagery helps in quantifying and understanding land cover changes. Because we aim to discover wetland change trends, time-series data have the potential for correlating and finding differences in adjacent periods between observations. Remotely sensed data from Landsat satellites are the most common source of optimal ground resolution and spectral bands for use in tracking and documenting land features. Owing to their long history and accessibility, Landsat satellites have often been used in land cover monitoring applications [14–16], allowing the use of historical data to track past adjacent change trends. Because remote sensing time series data are numerous, substantial computational power is needed for analysis. Remote sensing time series data are so numerous as to be infeasible to download, analyze, and manage using our computer power [17], with processing method and classification technology only raising the technological requirements. In this study, we used the Google Earth Engine (GEE), a cloud computing platform that can process and run such time-series data, enabling users to analyze petabytes of data on the fly without navigating the complexities of cloud-based parallelization (https://earthengine.google.com accessed on 5 February 2020). Google Earth Engine can be used for mapping [18] and classical algorithms, such as support vector machine (SVM), random forest (RF), classification and regression trees (CART), which can be freely implemented in GEE [17,19–21].

Previous studies such as [8,11,22] used satellite imagery to identify and map wetlands. Landsat imagery, owing to its time scales, spectral bands, and spatial resolution, holds great potential for wetland studies [2]. The consequences of climatic change and human pressure due to population growth are the main cause for the degradation of wetland ecosystems. The human activity caused by tremendous agricultural encroachment is the main driver of natural wetland degradation [23,24]. Wetland loss in the eastern regions of China (Northeast China, North China, Southeast China, and South China) is affected mainly by urbanization [25].

Since wetlands' resources are one of the most important natural environment resources in the world; understanding, protecting, and using them wisely are essential to attaining sustainable development. However, they remain among the world's least understood and most seriously abused assets [1]. The research gap and limited information throughout the world about wetlands have been revealed by current scientific reviews and studies. In China, we encountered little concern about wetland ecosystem resources and limited databases about long-term wetland change trends in the country, particularly in Guangling County. The wetland change trend data were sadly lacking. Wetland distribution and documentation have been very limited in the county. This paper uncovers wetland change trends over the past 3.7 decades and their links to climate variabilities and economic growth in the county to maintain and restore ecologically vulnerable areas for sustainable development of wetland ecosystems.

The specific objectives of the present study are to: (I) evaluate the performance of the five classical machine learning algorithms on time-series data classifications based on Google Earth Engine (GEE); (II) detect the change trends of wetland over the past decades in Guangling County, Shanxi Province, China; and (III) explore wetland change trends' link with the economic growth of the county and climate fluctuations in the area.

## 2. Materials and Methods

### 2.1. Study Area

The region of interest is Guangling County, in the northeast portal of Shanxi Province, China. Geographically, located between 113°57′4.75″~114°16′58″ E longitude and 39°37′50.74″~39°45′37″ N latitude. Guangling County (Figure 1) is surrounded by the Taihang Mountains and is adjacent to Yu County in Hebei Province in the east, Ling Qiu County in the south, Hun Yuan County in the west, and Yang Yuan County in the north. It comprises around 1225 km² of land area and a population of 185,000, of whom 150,000 are based on agricultural activities. Economic data at the county level show that the gross domestic product (GDP) is increasing continuously, reaching >3.985 billion yuan in 2019, a year-over-year increase of >8.8%.

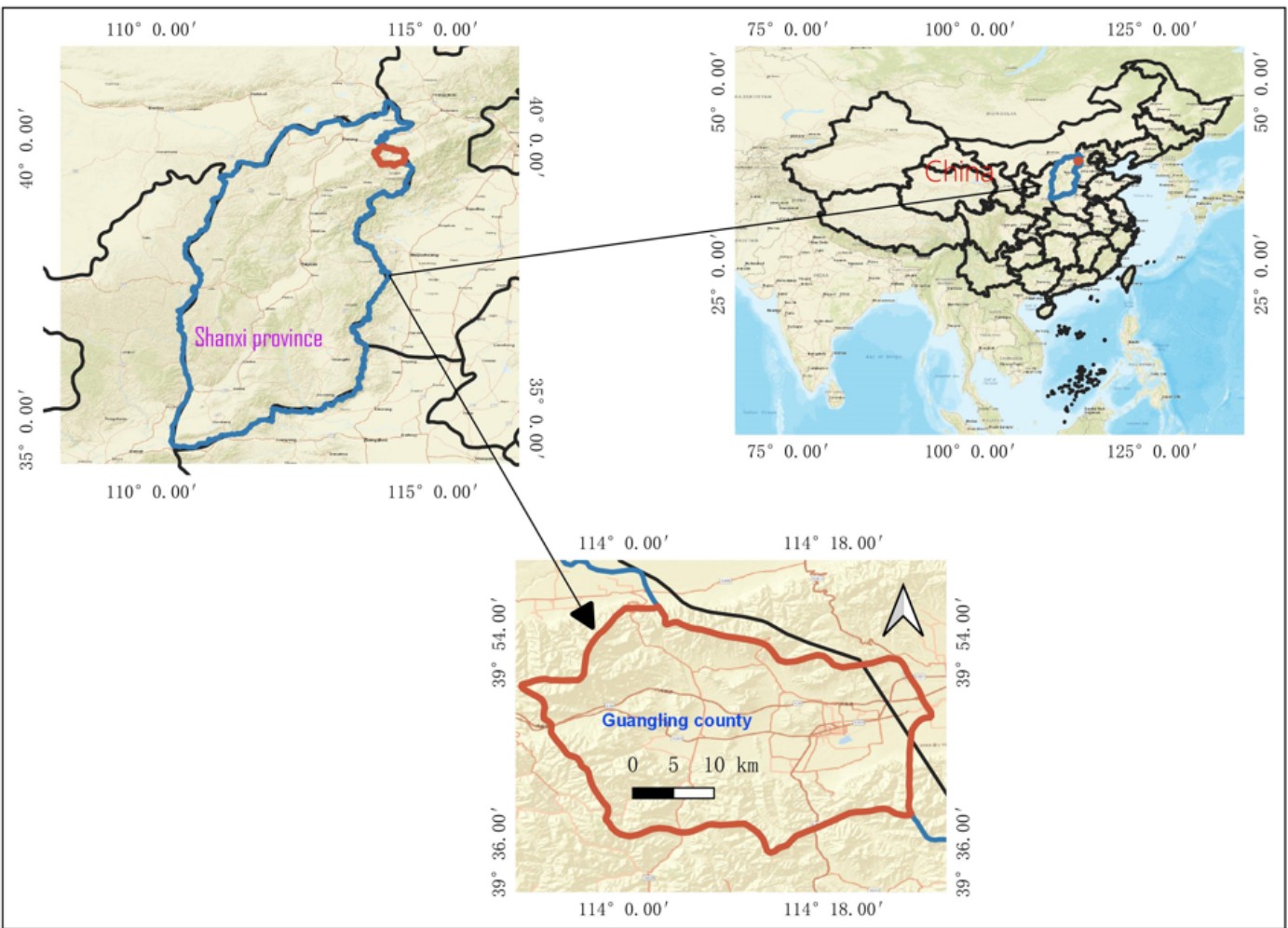

**Figure 1.** Location of Guangling County (the background is from Esri street map (https://www.arcgis.com/apps/mapviewer accessed on 1 October 2021)).

The study area has a continental monsoon climate, with significant annual variations in temperature (in meteorology, diurnal temperature variation is the variation between a high temperature and a low temperature on the same day, averaging 13.2 °C annually) and

changes in precipitation (with more than three-quarters of annual precipitation occurring from June to September and almost none in winter), and steppe climate refers to the climate of a region receiving precipitation below potential evapotranspiration (http://www.sx-guangling.gov.cn/ accessed on 1 September 2021). Guangling County is characterized by a temperate continental monsoon climate, with an extreme maximum temperature of 38.2 °C, a minimum temperature of −34 °C, an average annual temperature of 7 °C, a frost-free period of 134 days, precipitation of 388 mm, and 340 days of weather above grade II. The county has a weak industrial base and is well known for traditional agricultural activities. It is a source of more than 10 kinds of proven mineral resources, including coal, iron, germanium, gallium, manganese, limestone, granite, and particularly reserves of high-calcium limestone, high-grade magnesium-rich dolomite, and iron ore. The area is characterized by different geomorphic types mainly mountains, hills, and basins. Loess is widely distributed, and the terrain is high in the west and low in the east. Yongding River, the source of the Liuhu River, runs through the east and west and belongs to the Haihe River system. The county's lowest point is about 930 m above sea level, where the Huliu River flows out of the county.

### 2.2. Remote Sensing Datasets and Data Processing

When monitoring land use land cover change trends, time-series multitemporal and multisource remotely sensed data are essential. In this study, open-access satellite data based on compatible spatial resolutions were used. Being freely accessible and offering good spatial resolution, Landsat data are consistent with data from former missions, allowing assessment of long-term regional and global land cover changes [26]. In our experiment, data with high quality from Landsat 5, 7, and 8 are used for classification and prediction. For classifications, bands and periods of satellite imagery were selected. Owing to interference and distortion during the cloudy or rainy seasons, periods of clear weather with fewer cloudy and qualified images were used. We tried to use all cloudless freely available qualified Landsat imagery in GEE. Data sources are summarized in Table 1.

**Table 1.** Data sets list and descriptions used in our experiments.

| Datasets | Descriptions & Acquisition Date | Derived Variables | Sources |
|---|---|---|---|
| Remote sensing data: Level-1 Landsat images | 167 Landsat images—30 m resolution (2 TM images in 1984, 14 TM images in 1990, 24 TM images in 1995, 21 ETM+ images in 2000, 33 ETM+ images in 2005, 2 TM images in 2010, 19 OLI images in 2015, 52 OLI images in 2020) | Normalized Difference Water Index (NDWI), Normalized Difference Vegetation Index (NDVI), Modified Normalized Difference Water Index (MNDWI), Normalized Difference Built-up Index (NDBI), Bare Soil Index (BSI) | USGS https://landsat.usgs.gov/ accessed on 4 April 2021 in GEE |
| Ground truth data | Field data using GPS recorders (28–30 August 2017) | 35 filed observations | Field trip |
| Shuttle Radar Topography Mission (SRTM) | Digital elevation data (11–22 February 2000) | Elevation and Slope | https://cmr.earthdata.nasa.gov/search/concepts/C1000000240-LPDAAC_ECS.html accessed on 15 April 2021 |
| Google Earth | Time series imageries | Wetland–non-wetland | Google Inc. |
| Open street map | Open-source data | Vector polygon land cover features | https://www.openstreetmap.org/ accessed on 1 May 2021 |
| Global surface water | Distributions of water surface (1 March 1984–31 December 2020) | Waterbody | https://global-surface-water.appspot.com/# accessed on 1 May 2021 |
| Climate | Temperature, precipitation, and relative humidity (1981–2020) | Climate time-series data | https://power.larc.nasa.gov/data-access-viewer/ accessed on 10 June 2021 |

Because Tier 1 (T1) datasets available in GEE were used, no further preprocessing was required. The data met preprocessing quality requirements, but the imagery covering our study area was not clear enough (too bright pixels and black pixels), so we applied to preprocess. The study area is surrounded by hilly and mountainous regions, remotely sensed images may have a high probability of being distorted by shadows [20]. Cloud masking metric was applied in preprocessing stage. For cloud masking (with deselecting images having more than 3% of cloud shadow and images having more than 5% of cloud cover), the metrics median () and mosaic () written in JavaScript were applied in GEE. Thus, the cFmask, cloudScore, cloud, and cloud shadow masking metrics written in JavaScript were applied to image composites of Landsat Tier 1 to remove various types of clouds and noises to produce a per-pixel, minimally cloudy or cloud-free, and noise-free multispectral composite of the region of interest.

As Table 1 specifies, spectral indices from optical imagery such as *NDWI*, *NDVI*, *NDBI*, and *BSI* were extracted and used for feature extraction and model inputs in classification. The Normalized Difference Water Index (*NDWI*) is a spectral index used for detecting inundation and separating dry land from water bodies. *NDWI* was formulated from spectral wavelengths of near-infrared *NIR* and short-wave infrared *SWIR* channels, which are sensitive to open water and vegetation liquid water [27] (Equation (1)). Thus, *NDWI* was used to clearly show water bodies and distinguish wet areas from dry land. The Normalized Difference Vegetation Index (*NDVI*), the most common spectral index in land cover monitoring applications, was used for vegetation studies because of its sensitivity to photosynthetically active biomass and phonological dynamics in vegetation or forest [28] (Equation (2)). The Normalized Difference Built-up Index (*NDBI*). Urban areas or built-up areas and barren land show drastically different reflectance between *NIR* and short-wave infrared *SWIR*, whereas vegetation has a slightly larger or smaller DN value on *SWIR* than on *NIR*, so we used *NDBI* for verification (Equation (3)). The Bare Soil Index (*BSI*) was calculated by combining blue, red, near-infrared, and short-wave infrared (*SWIR*) spectral bands (Equation (4)). *BSI* can be used in numerous remote sensing applications, including soil mapping and crop identification (in combination with *NDVI*) [29].

$$NDWI = \frac{NIR - SWIR}{NIR + SWIR} \tag{1}$$

$$NDVI = \frac{NIR - Red}{NIR + Red} \tag{2}$$

$$NDBI = \frac{SWIR - NIR}{SWIR + NIR} \tag{3}$$

$$BSI = \frac{(SWIR + Red) - (Green + Blue)}{(SWIR + Red) + (Green + Blue)} \tag{4}$$

In Landsat 4–7, bands 4 NIR and 3 Red were used to calculate *NDVI*; Bands 4 *NIR* and 5 *SWIR* to calculate *NDWI* [30]; Bands 5 *SWIR* and 4 *NIR* to calculate *NDBI*; and Bands 5, 3, 2, and 1 *SWIR*, Red, Green, and Blue, respectively, to calculate *BSI*. In Landsat 8, bands 5 *NIR* and 4 Red were used to calculate *NDVI*; Bands 5 *NIR* and 6 *SWIR* to calculate *NDWI*; Bands 6 *SWIR* and 5 *NIR* to calculate *NDBI* [31]; and Bands 6, 4, 3, and 2 *SWIR*, Red, Green, and Blue, respectively, to calculate *BSI*. Cloud-masked Landsat imagery was used to generate a per-pixel median composite of each of the multispectral bands and the spectral indices. Bands 1 Blue to 5 *SWIR* in Landsat 5 and 7, Bands 2 Blue to 7 *SWIR*2 in OLI, and Equations (1)–(4) *NDVI*, *NDWI*, *NDBI*, and *BSI* were extracted from the median composite, clipped to the region of interest, and finally used as model inputs.

### 2.3. Sample Data

Adequate datasets are key to training machine learning models for use with remotely sensed image analyses. The reference data used for dataset collection were field data as ground truth data, obtained by field survey on 28–30 August 2017. The field data were

recollected and merged into three classes, as described in Section 2.4, with Open Street Map (OSM) and historical data of high-resolution Google Earth imagery used to collect datasets (evaluation data) incorporating ground truth data. To precisely acquire wetlands of the study area, we used various data sources to collect datasets, including Landsat images, Shuttle Radar Topography Mission (SRTM), and Global Surface Water (GSW) [32]. A total of 6792 data were collected, of which 5335 pixels or points were used as training data and 1457 points as validation or test data. The main data used for classifications and predictions were Landsat images, including TM, ETM+, and OLI imagery. All imagery used was the least cloud cover or noncloudiness imagery from early 1984 (Landsat TM) to the end of 2020 (Landsat OLI) at a 5-year interval.

### 2.4. Classification System

We performed image classification using pixel-based image analysis approaches, which are more convenient for images with heterogeneity in nature than for object-based image analysis [33]. Wetlands could include areas of marsh, fen, peatland, and water, whether natural or artificial, permanent or temporary, whether water is static or flowing, fresh, brackish, or salty—including areas of marine water with a low-tide depth of not more than 6 m. Based on this definition, considering the features of global wetlands, in this study, water both natural and artificial rivers or reservoirs, was considered as wetland [34]. All mapped land cover classifications and land cover changes were categorized as wetland or non-wetlands. Thus, our experiment attempted to classify land cover features of the study area as wetland (including forested wetlands) and non-wetland (forest and considering all other classes in one class); Figure 2 shows classification system (e.g., the first level of Anderson [35,36]) and Figure 3 is general workflow of the study implemented for proposed wetland mapping.

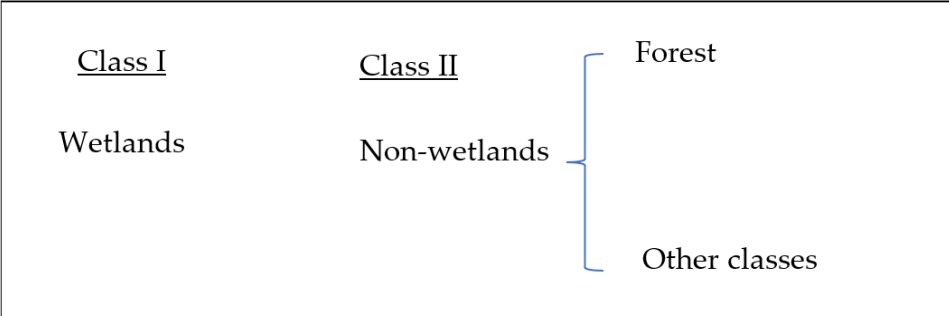

**Figure 2.** Classification system for remote sensing data from the first level of the Anderson classification system.

### 2.5. Machine Learning Classifiers in GEE

Machine learning (ML) classifiers have emerged as a field of artificial intelligence that uses complex reference data to build a classifier based on data-driven decisions [30]. We implemented land cover classifications using algorithms CART, Gradient tree boost, Minimum distance, RF, and SVM.

#### 2.5.1. Classification and Regression Trees (CART)

CART [37] is a decision tree-based model that uses tree structures for simplicity and to reduce sample dimensions. The tree-structured approach is simpler in regression than in classification. CART, a predictive modeling approach employed in machine learning and other fields, uses a decision tree as a predictive model for use with tree models where the target variables can take a discrete set of values known as classification trees and decision trees where the target variables can take continuous values is regression trees.

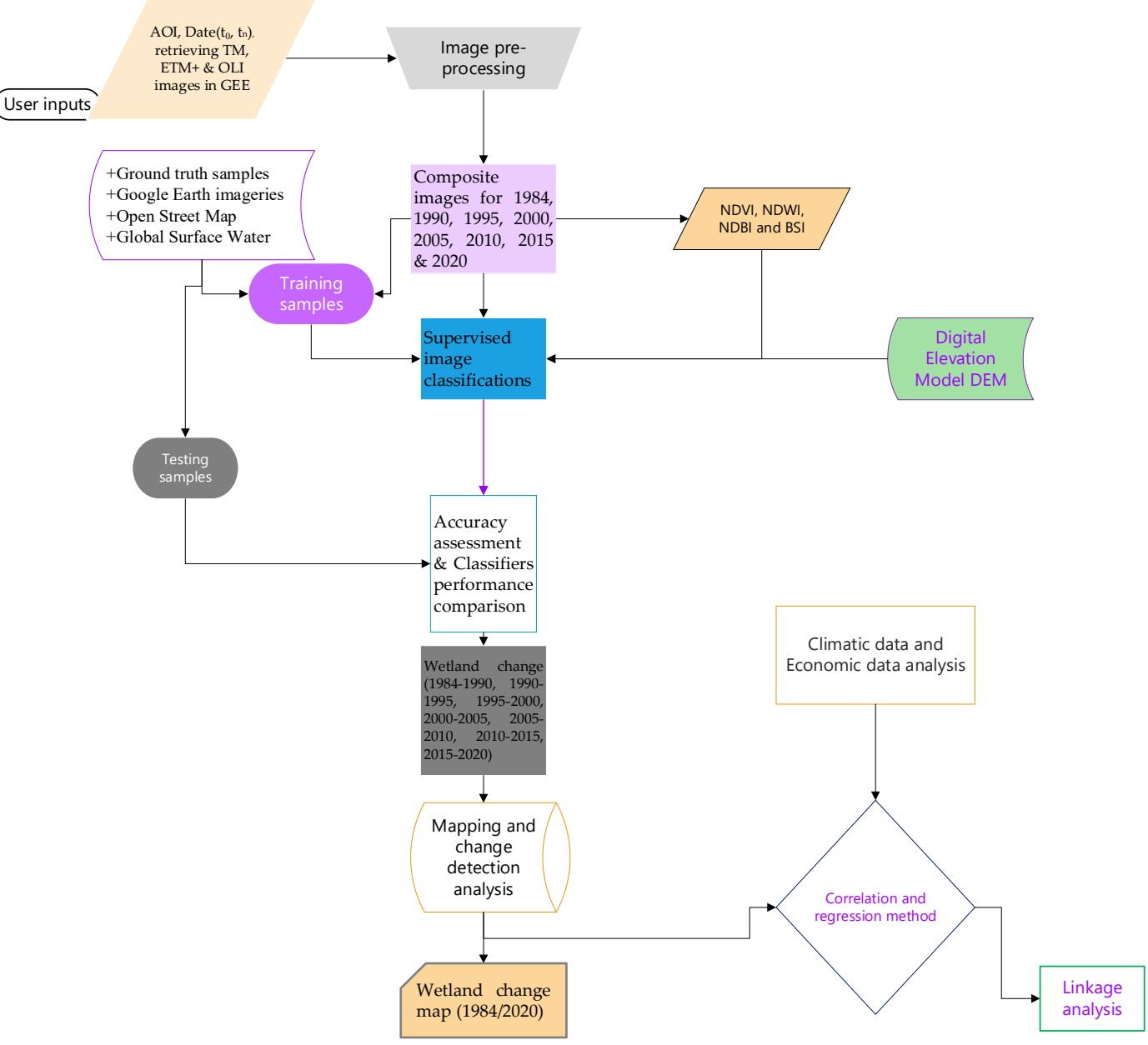

**Figure 3.** The workflow of the proposed wetland mapping.

### 2.5.2. Gradient Tree Boost

The concept of the gradient boosting machine originated with Leo Breiman [38]. Subsequently, gradient boosting algorithms were developed by J.H. Friedman [39]. When the decision tree is a weak learner, the algorithm is gradient-boosting and is typically used with CART. Modification of Friedman's design into gradient boosting algorithms improves the quality of fit for each base learner.

The author in [39] let gradient tree boosting at the $m - th$ step fit a decision tree to pseudo residuals output of the $h_m(x)$ for input $x$ is

$$h_m(x) = \sum_{j=1}^{J_m} b_{jm} 1_{R_{jm}}(x) \tag{5}$$

The modified model by Friedman into the "tree boost", where $F_m(x)$ is probability estimation:

$$F_m(x) = F_{m-1}(x) + \sum_{j=1}^{J_m} \gamma_m 1_{R_{jm}}(x), \ \gamma_{jm} argmin_\gamma \sum_{x_i \epsilon R_{jm}} L(y_i, the \ F_{m-1}(x_i) + \gamma) \quad (6)$$

where $x_i$ is input variables, $y_i$ is pseudo-responses, $L$ is loss function, $m$ is iteration step, $J_m$ is the number of its leaves, the partitions the input space into $J_m$ disjoint regions, $R_{1m}, \ldots, R_{J_m}$ predicts a constant value in each region, and $b_{jm}$ is the value predicted in the region $R_{jm}$.

### 2.5.3. Minimum Distance

In minimum distance classifiers, the items classified are groups of measurement vectors, all from samples rather than individual vectors as they would be in more conventional vector classifiers. Specifically, the sample, which is to say the group of vectors, is classified into the class whose known or estimated distribution most closely resembles the estimated distribution of the sample to be classified. The measure of resemblance is a distance measure in the space of distribution functions [40], meaning that the minimum distance classifier can classify unknown image data into classes that minimize the distance between the image data and the class in multifeatured space, with the distance representing the index of similarity. In minimum distance algorithm, based on the training datasets to find a mean value of pixels. This classifier can find the minimum distance from the mean values of each class of the training datasets to the digital value of each pixel in imagery.

### 2.5.4. Random Forest (RF)

RF has a combination of tree predictors such that each tree depends on the values of a random vector sampled independently, with the same distribution for all trees in the forest [41]. It is the most common classifier used in land cover classification because of its classification accuracy. In our experiment, 10 decision trees were used to achieve better classification accuracy. According to [41], given an ensemble of classifiers, $h_1(x), h_2(x), \ldots, h_k(x)$ and with the training set drawn at random from the distribution of the random vector $Y, X$, the margin function is defined as:

$$mg(X, Y) = av_k I(h_k(X) = Y) - \max_{j \neq Y} av_k I(h_k(X) = j). \quad (7)$$

where $I()$ is the indicator function.

The generalization error is calculated by:

$$PE^* = p_{X,Y}(mg(X, Y) < 0) \quad (8)$$

The subscripts $X, Y$ indicates that the probability is over the $X, Y$ space. The margin function measures the extent to which the average number of votes at $X, Y$ for the right class exceeds the average vote for any other class: the larger the margin, the more accurate the classification.

$$RF, \ h_k(X) = h(X, \Theta_k).$$

As the number of trees increases, the sequences of $\Theta_1 \ldots. PE^*$ converge to

$$P_{X,Y}(P_\Theta(h(X, \Theta) = Y - \max_{j \neq Y} P_\Theta(h(X, \Theta) = j) < 0 \quad (9)$$

Thus, RF does not overfit as the number of trees increases.

### 2.5.5. Support Vector Machine (SVM)

To be effectively functional and produce reasonable classification accuracy, SVMs should address using certain parameters. The radial basis function (RBF) kernel was used. Unlike the linear kernel, it maps nonlinear samples into a higher-dimensional space, so

it can handle cases when the relation between class labels and attributes is nonlinear. In training datasets, each instance contains one target value (i.e., the class labels) and several attributes (i.e., the features of observed variables) [42]. The common goal of SVM is to produce a model based on the training data that predicts the target values of the test data given only the test data attributes.

With a certain training set of instances–label pairs $(X_i, Y_i)$, $i = 1 \ldots \ldots l$ where $X_i \in \mathbb{R}^n$ and $y \in \{1, -1\}^l$ an SVM requires the solution of limited optimization problems [42–44]

$$\min_{wb\xi} \frac{1}{2} W^T W + C \sum_{i=1}^{l} \xi i$$

Subject to

$$y_i(W^T \varphi(X_i) + b\,() \geq 1 - \xi i \, , \, \xi i \geq 0 \tag{10}$$

where $X_i$ is a training vector, mapped into a higher-dimensional space by the function $\varphi$, and $C > 0$ is the penalty parameter of the error term. SVM finds a linear separating hyperplane with the maximal margin in this higher-dimensional space, RBF kernel:

$$K(X_i, X_j) = \exp\left(-\gamma||X_i - X_j||^2\right), \, \gamma > 0. \tag{11}$$

where $K(X_i, Y_i) = \varphi\,(X_i)^T \varphi(X_j)$ is the kernel function. $\gamma$, $T$ are the kernel parameters.

*2.6. Classification Accuracy Assessment*

After data classification, especially of satellite imagery in remote sensing, the classifier's model performance should be verified and evaluated to assess the performance of the classifier's model. The most common way of assessing classification accuracy is to calculate the error matrix of the classification, which is regarded as confusion of the matrix of the classifier's model. The confusion matrix is used to compare reference data and the corresponding result of the classification. From the confusion matrix, the overall accuracy of the model, kappa statistics, user accuracy, producer accuracy, omission error, and commission error can be calculated. In our experiment, we assessed classification accuracy in GEE, assessing the model's overall accuracy (dividing the total number of classified pixels by the total number of reference pixels), kappa statistics (Equation (12)), user accuracy (number of correctly classified pixels in each class per total row for that class), and producer accuracy (number of correctly classified pixels in each class per total column for that class).

$$y = (N\sum_{i=1}^{r} x_{ii} - \sum_{i=1}^{r}(x_{i+} \cdot x_{+i}))/(N^2 - \sum_{i=1}^{r}(x_{i+} \cdot x_{+i})) \tag{12}$$

where $N$ is total number of observations in the confusion matrix, $r$ the number of rows in the confusion matrix, $x_{ii}$ the number of observations in a row $i$ and column $i$, $x_{i+}$ the total number of observations in a row $i$, and $x_{+i}$ the total number of observations in the column $i$.

*2.7. Grey Correlation Analysis*

Climate change calls for use of scientific techniques to analyze trends and make decisions. In this study, Grey's relational analysis GRA [45] was used to analyze correlations between climate change and wetland trends. GRA allows us to determine an appropriate solution for real-world problems, pioneered by Chinese professor Deng Ju long. One of the most widely used models of Grey system theory, it is promoted by various scholars and institutions, such as the International Journal of Grey Systems and the International Association of Grey Systems and Decision Sciences (IAGSUA). Grey correlation analysis is used to

determine correlation coefficients and degrees between observations. The comprehensive factors of climate change's impact on wetlands exhibit a strong Grey character, which is to say incomplete information and uncertainty [46], so Grey system theory is suitable for comprehensively evaluating the correlation between climate variables and wetlands. Grey relational analysis was used to evaluate the relationship between land changes such as wetlands and environmental factors [46,47].

$$r(x_o, x_i)\frac{1}{n} \sum_{k=1}^{n} r(x_0(k), x_i(k))$$

$$r(x_0(k), x_i(k)) = \frac{\min_i \min_k |x_0(k) - x_i(k)| + \varrho \max_i \max_k |x_0(k) - x_i(k)|}{|x_0(k) - x_i(k)| + \varrho \max_i \max_k |x_0(k) - x_i(k)|} \tag{13}$$

where $x_o$ is the reference data, $x_i$ is the comparison sequence, "$\varrho$" is the resolution coefficient, and "$\varrho$" $\in [0, 1]$, $x_0(k)$, and $x_i(k)$ the number of points $k$ of $x_0$ and $x_i$, respectively. To find their degree of Grey correlation, we used wetland area as $x_0$; climate data, including temperature, humidity, and precipitation as $x_i$, and "$\varrho$" = 0.5 as the related effective study [46,47].

## 3. Results

### 3.1. Classifications, Change Detection and Algorithms' Model Performance

3.1.1. Classifications

In our proposed system, land cover classifications were performed aimed to detect wetlands. Land cover classifications from 1984 to 2020 were accomplished. Landsat imagery was offered for those periods, so time series satellite images from the Landsat Thematic Mapper (TM) to Operational Land Imager (OLI) were investigated and used for land cover classifications. Enhanced images were obtained and used for the calculation of spectral indices (Section 2.2). Classifications were made within an interval of 5 years, eight land cover classification and land cover change thematic maps were obtained (Figures 4 and 5).

3.1.2. Accuracy Assessment

Classified maps of 1984, 1990, 1995, and 2010 were created using satellite imagery from Landsat 5 (Thematic Mapper). Accuracy for each year was calculated based on the confusion matrix, assessing the model's overall accuracy, kappa statistics, producer accuracy, and user accuracy. For instance, the classification of 2015 showed wetland class results with the highest user accuracy of 100% and producer accuracy of 100%. Forest classification produced 97.28% user accuracy and 95.29% producer accuracy, and other classifications produced 95.53% user accuracy and 97.43% producer accuracy. Overall classification accuracy was 97.53%, with a kappa statistic of 0.963. The following tables show the error matrix obtained by SVM classifier: Table 2 from classification data of 1984; Table 3 from classification data of 2000; Table 4 from classification data of 2005, Table 5 from classification data of 2020.

**Table 2.** Error matrix obtained by SVM (1984).

| | | Wetland | Forest | Others | Row Total | User's Accuracy |
|---|---|---|---|---|---|---|
| Classification Data | Wetland | 457 | 0 | 7 | 464 | 0.985 |
| | Forest | 0 | 458 | 20 | 478 | 0.958 |
| | Others | 0 | 24 | 491 | 515 | 0.953 |
| | Column total | 457 | 482 | 518 | 1457 | |
| | Producer's accuracy | 1.00 | 0.950 | 0.950 | | |
| Overall Accuracy | | 0.965 | | | | |
| Kappa statistics | | 0.947 | | | | |

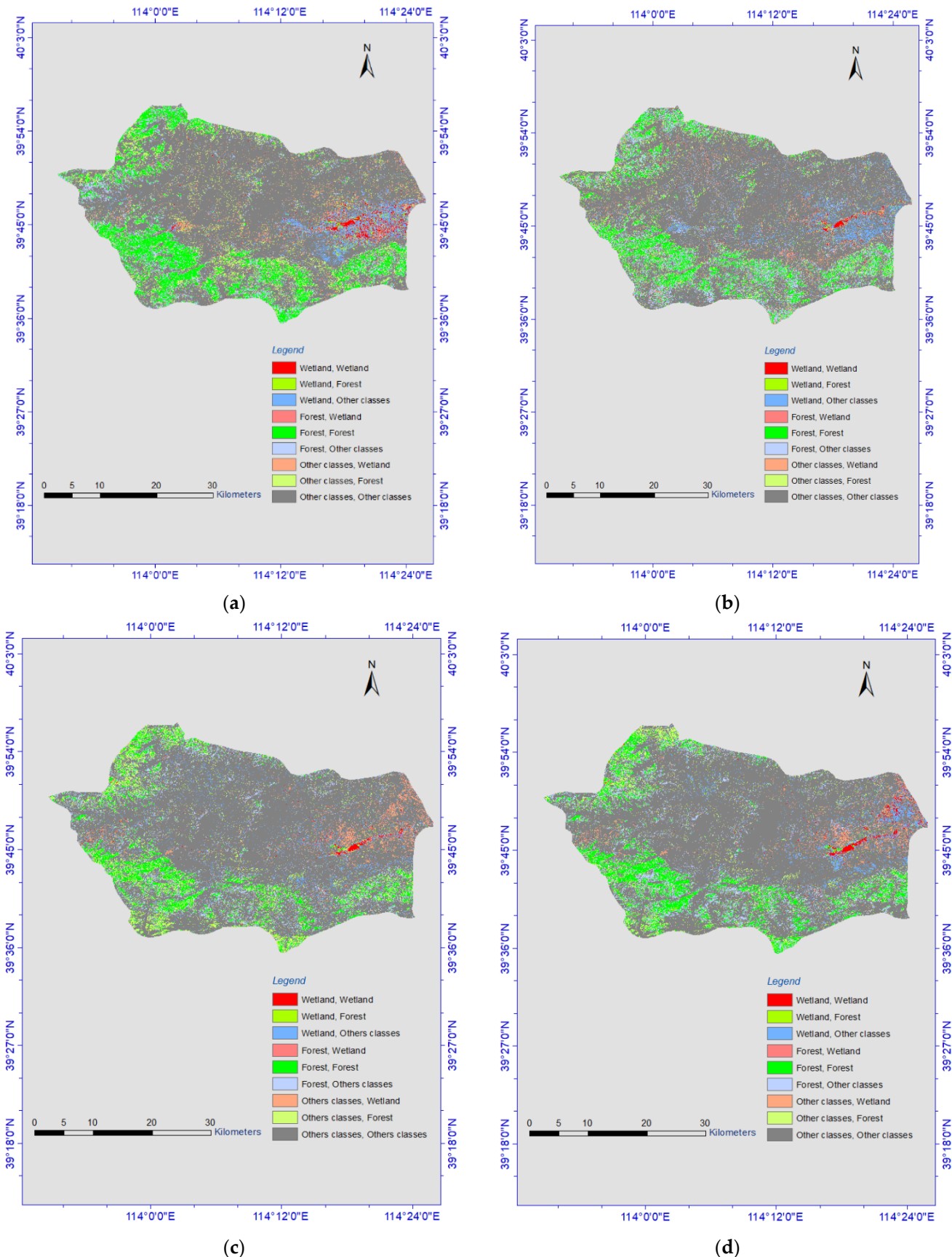

**Figure 4.** *Cont.*

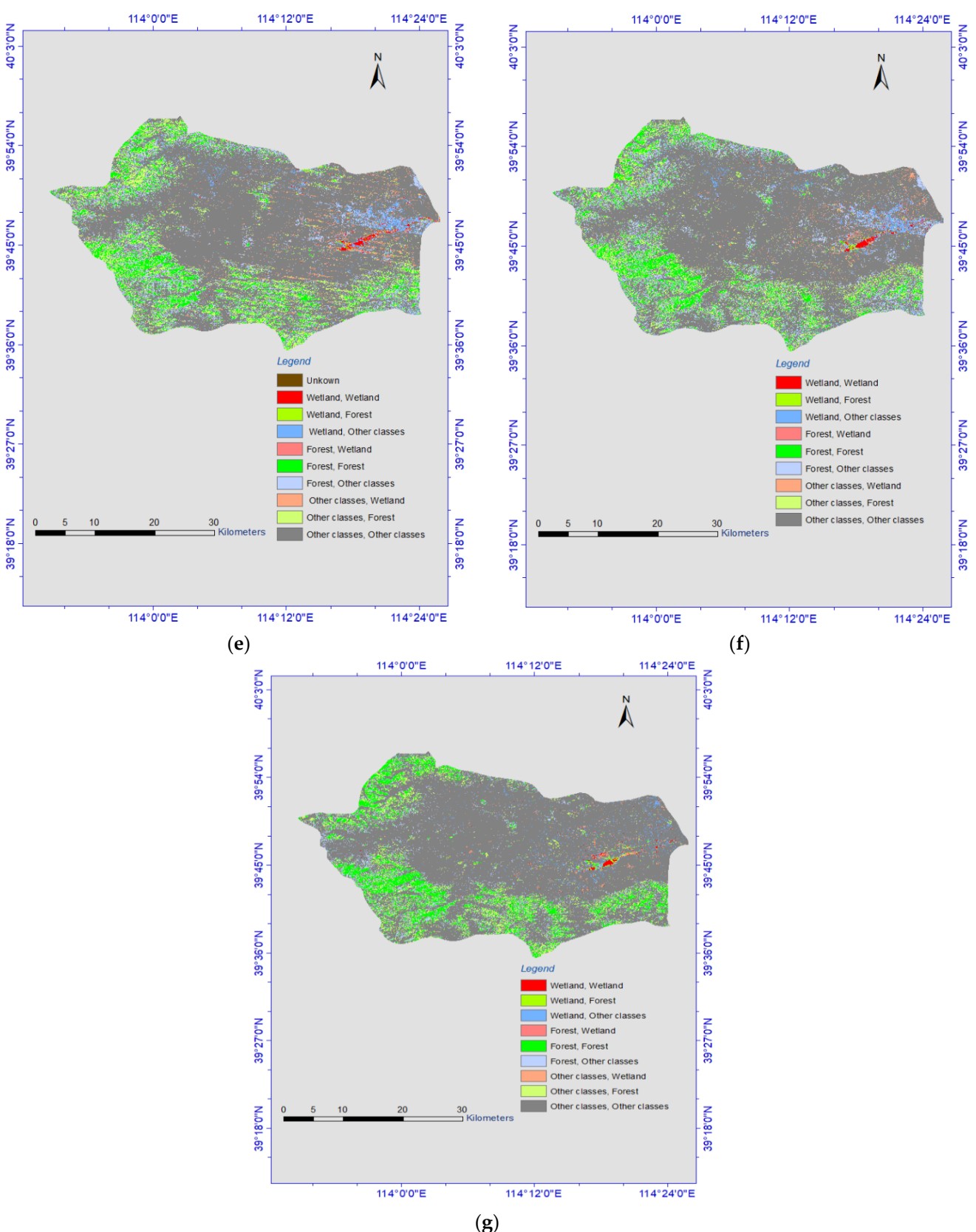

**Figure 4.** Land cover change maps: (**a**) 1984–1990, (**b**) 1990–1995, (**c**) 1995–2000, (**d**) 2000–2005, (**e**) 2005–2010, (**f**) 2010–2015, (**g**) 2015–2020.

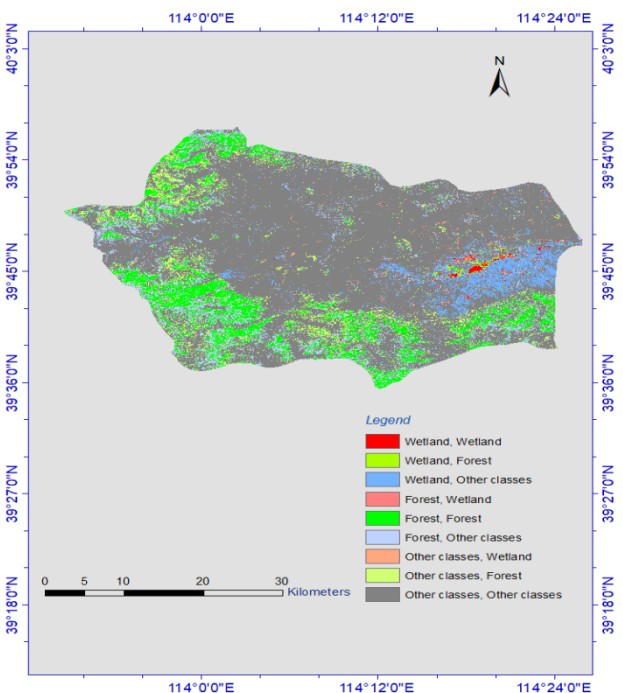

**Figure 5.** Thematic map of overall land cover change detection, 1984–2020.

**Table 3.** Error matrix obtained by SVM (2000).

| | | Wetland | Forest | Others | Row Total | User's Accuracy |
|---|---|---|---|---|---|---|
| Classification Data | Wetland | 464 | 0 | 0 | 464 | 1.00 |
| | Forest | 56 | 365 | 57 | 478 | 0.764 |
| | Others | 1 | 7 | 507 | 515 | 0.984 |
| | Column total | 521 | 372 | 564 | 1457 | |
| | Producer's accuracy | 0.891 | 0.981 | 0.899 | | |
| Overall Accuracy | | 0.917 | | | | |
| Kappa statistics | | 0.875 | | | | |

**Table 4.** Error matrix obtained by SVM (2005).

| | | Wetland | Forest | Others | Row total | User's Accuracy |
|---|---|---|---|---|---|---|
| Classification Data | Wetland | 464 | 0 | 0 | 464 | 1.00 |
| | Forest | 22 | 332 | 124 | 478 | 0.695 |
| | Others | 0 | 58 | 457 | 515 | 0.887 |
| | Column total | 486 | 390 | 581 | 1457 | |
| | Producer's accuracy | 0.955 | 0.851 | 0.787 | | |
| Overall Accuracy | | 0.860 | | | | |
| Kappa statistics | | 0.789 | | | | |

**Table 5.** Error matrix obtained by SVM (2020).

| | Class name | Wetland | Forest | Others | Row Total | User's Accuracy |
|---|---|---|---|---|---|---|
| Classification Data | Wetland | 464 | 0 | 0 | 464 | 1.00 |
| | Forest | 0 | 469 | 9 | 478 | 0.981 |
| | Others | 14 | 4 | 497 | 515 | 0.965 |
| | Column total | 478 | 473 | 506 | 1457 | |
| | Producer's accuracy | 0.971 | 0.992 | 0.982 | | |
| Overall Accuracy | | 0.981 | | | | |
| Kappa statistics | | 0.960 | | | | |

As shown in Tables 4 and 5, classifications for 2000 and 2005 were performed using satellite imagery from Landsat 7 (Enhanced Thematic Mapper plus ETM+).

By contrast, classifications for 2015 and 2020 were performed using data from Landsat 8 (Operational Land Imager), assessing the model's accuracy based on the test data.

Classifications were performed using five classical machine learning algorithms: SVM, RF, gradient tree boost, minimum distance, and CART. Model performance was evaluated through accuracy assessment based on testing datasets (summarized in Table 6).

**Table 6.** Summary of classification accuracy assessment achieved by classifiers (2015).

| Classifiers | Overall Accuracy (%) | Kappa Statistics |
|---|---|---|
| Support Vector Machine SVM | 97.53 | 0.963 |
| Random Forest RF | 97.30 | 0.960 |
| Gradient tree boost | 96.91 | 0.954 |
| Minimum distance | 84.49 | 0.768 |
| Classification and Regression Trees CART | 96.50 | 0.947 |

### 3.2. Wetland Change Detection

The accurate change detection of land cover's features is critical for understanding the differences over time and to know the relationships between human activities and natural phenomena. To distinguish the long-term and spatiotemporal dynamics of wetlands, change detection is required [48]. Post classification process, change detection was implemented in GEE based on the land cover classification results of time series data. Because knowing how many pixels are converted to other classes can decide the spatial distribution of wetland resources and their changes over long adjacent periods, GEE can be used for simple tasks, such as creating composites of satellite imagery, but can also handle more complex tasks, such as detecting long-term dynamics of things. Based on performance evaluations of the five algorithms implemented on GEE, we chose SVM to detect changes in classes for each year, because it offered better classification accuracy than the other classifiers. Change detection was performed from 1984 to 2020 at 5-year intervals. Figures 3 and 4 show the eight land cover change maps generated, (1984–1990, 1990–1995, 1995–2000, 2000–2005, 2005–2010, 2010–2015, and 2015–2020).

Land cover change trends were obtained from the SVM-RBF model (Figures 4 and 5). As presented in Table 7, the change trends shown in Figure 4 are defined and calculated in terms of pixels change of each class.

Of all land classes change over the past 3.7 decades, Figure 5 and Table 8 reveal the critical land features change was observed from 1984 to 2020. Wetland conversion to other classes has undergone the greatest change and the maximum wetland loss recorded. As shown in Figure 6, the wetland change trends over the past 3.7 decades were identified in this study in terms of square kilometers.

### 3.3. Climate Variation and Wetland Change

The most common characteristics of weather or climatic variables include temperature, precipitation or rainfall, sunshine, wind, atmospheric pressure, and humidity. The values and fluctuations of these climatic variables can easily affect the spatial extent and distribution of wetlands. Precipitation, temperature, and humidity are the most common factors. We investigated the linkages between precipitations, temperature, and relative humidity with wetland change. The climatic time-series data correlated with the corresponding wetland changes of the area using a linear relationship.

**Table 7.** Pixel changes (**A**) 1984–1990, (**B**) 1990–1995, (**C**) 1995–2000, (**D**) 2000–2005, (**E**) 2005–2010, (**F**) 2010–2015, (**G**) 2015–2020.

| Class | Wetland | Forest | Others |
|---|---|---|---|
| | | (**A**) | |
| Wetland | 198,103.21 | 54,914.47 | 595,001.58 |
| Forest | 41,834.44 | 1,919,619.87 | 125,025.02 |
| Others | 515,025.02 | 1,177,495.84 | 10,202,487.88 |
| | | (**B**) | |
| Wetland | 71,568.34 | 35,780.00 | 647,614.33 |
| Forest | 75,958.75 | 1,312,292.20 | 1,763,779.22 |
| Others | 470,257.54 | 980,071.96 | 10,597,427.46 |
| | | (**C**) | |
| Wetland | 65,859.94 | 62,175.95 | 489,748.74 |
| Forest | 22,305.78 | 1,071,835.22 | 1,234,003.17 |
| Others | 436,639.12 | 1,148,719.44 | 11,423,462.44 |
| | | (**D**) | |
| Wetland | 70,485.16 | 38,984.21 | 415,243.47 |
| Forest | 108,147.45 | 1,082,655.27 | 1,089,067.29 |
| Others | 290,300.44 | 1,749,882.56 | 11,101,326.66 |
| | | (**E**) | |
| Wetland | 65,317.60 | 27,601.71 | 314,326.55 |
| Forest | 50,296.11 | 1,255,553.09 | 11,694,204.24 |
| Others | 329,650.38 | 1,554,975.75 | 10,662,553.35 |
| | | (**F**) | |
| Wetland | 56,170.67 | 30,223.07 | 320,864.13 |
| Forest | 31,033.80 | 1,177,405.57 | 1,791,838.08 |
| Others | 234,078.80 | 1,348,116.71 | 10,965,018.96 |
| | | (**G**) | |
| Wetland | 43,790.92 | 31,438.36 | 246,043.99 |
| Forest | 3528.18 | 1,488,849.63 | 1,063,367.54 |
| Others | 115,181.13 | 1,172,627.65 | 11,789,922.40 |

**Table 8.** The maximum number of pixel changes detected from one class to another class, 1984–2020.

| Class | Wetland | Forest | Others |
|---|---|---|---|
| Wetland | 56,550.87 | 36,990.8 | 754,477.59 |
| Forest | 9145.70 | 1,412,220.02 | 1,790,356.07 |
| Others | 110,201.13 | 901,114.02 | 10,883,693.59 |

Precipitation time-series data (Figure 7) for Guangling County were taken from meteorological data (1956–2020), and compared with time-series data calculated from freely available quasi global rainfall datasets; Climate Hazards Group InfraRed Precipitation with Station data (CHIRPS) was used in this study. The CHIRPS dataset builds on previous approaches by using "smart" interpolation techniques and high-resolution, long-term precipitation estimates based on infrared cold cloud duration (CCD) observations [49]. Figure 8 is showing the linear relation between inter-annual rainfall and the wetland area of Guangling County formulated from preceding year blocks of annual rainfall data corresponding to wetland distribution change in each of eight different years (1984, 1990, 1995, 2000, 2005, 2010, 2015, and 2020).

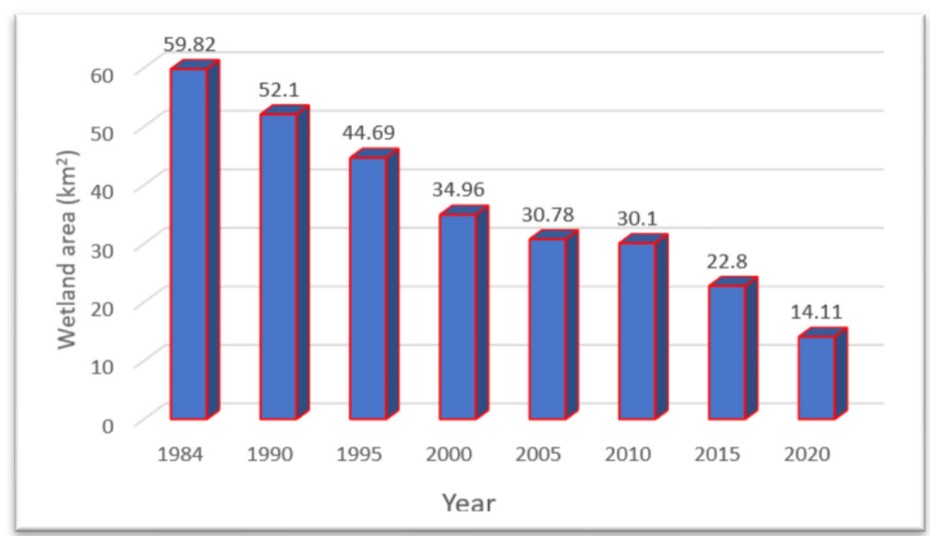

**Figure 6.** Wetland area coverage within Guangling County, 1984–2020.

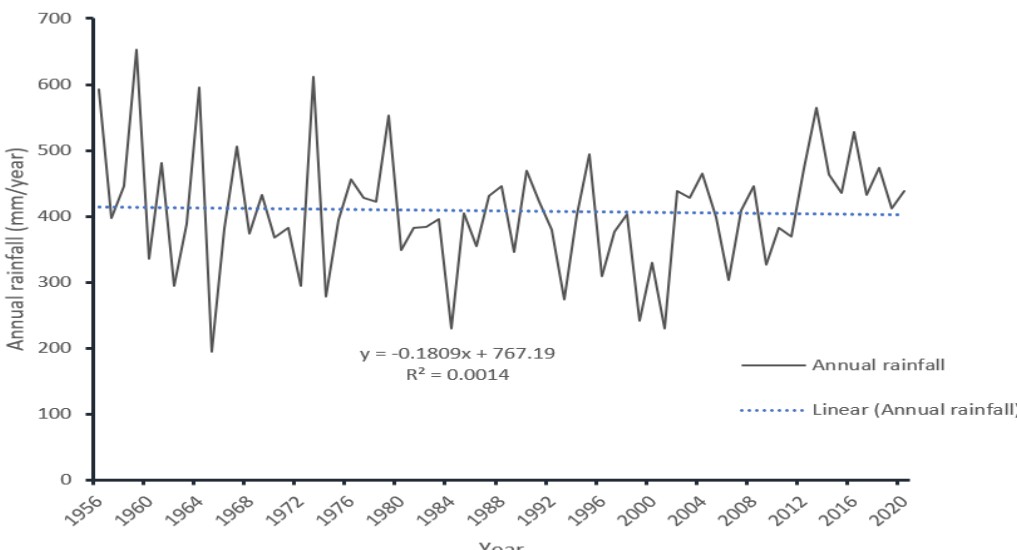

**Figure 7.** Annual rainfall of Guangling County, 1956–2020.

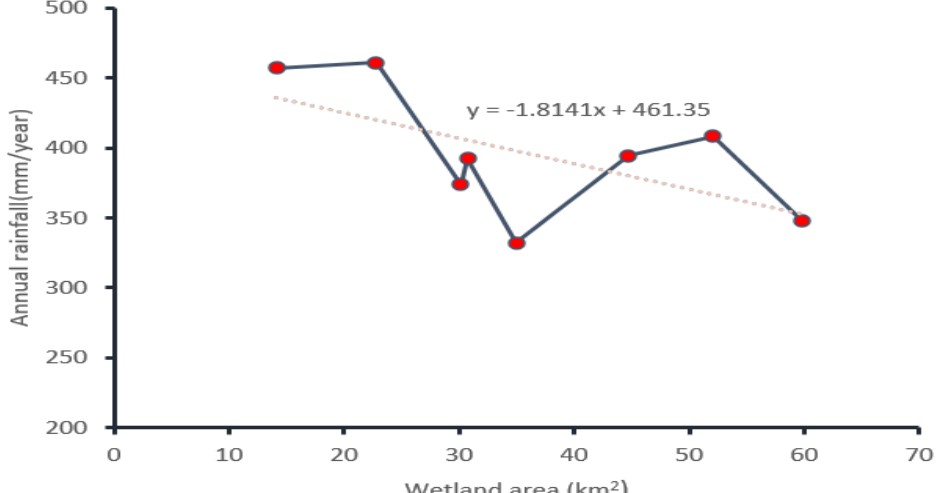

**Figure 8.** The linear relationship between annual rainfall and wetland change.

Temperature is the most influential variable, being strongly connected to the treatment of sustainable natural environments, particularly wetlands. As the current meteorological studies show climate variables are becoming one of the global problems, among them temperature is the most significant factor showing increments.

The highest increment of annual average temperature was approximately 2.4 °C (1995–2020), which had the greatest negative impact on wetland loss. Figure 9 shows the continuous variation trend of the annual average temperature of the study area, 1981–2020. The trendline reveals that the annual average temperature rises in Guangling County. In the past 3.7 decades, on average it was risen by 0.75 °C with the value of $R^2$ 14.24%.

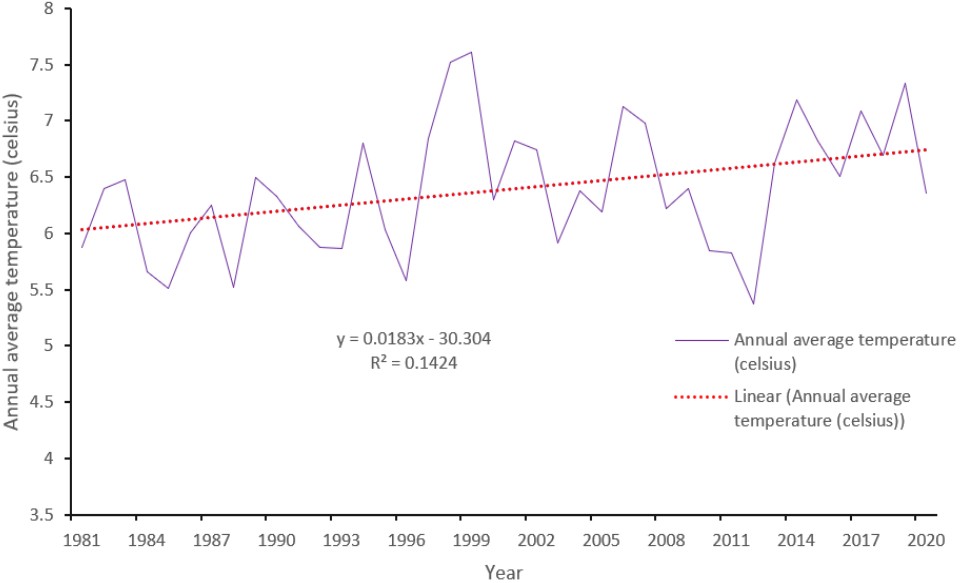

**Figure 9.** Annual average temperature of Guangling County, 1981–2020.

A linear relationship with wetland area shown in the figure below (Figure 10), which was calculated from the preceding year blocks of annual average temperature data corresponding to wetland distribution change in each of eight different years (1984, 1990, 1995, 2000, 2005, 2010, 2015, and 2020).

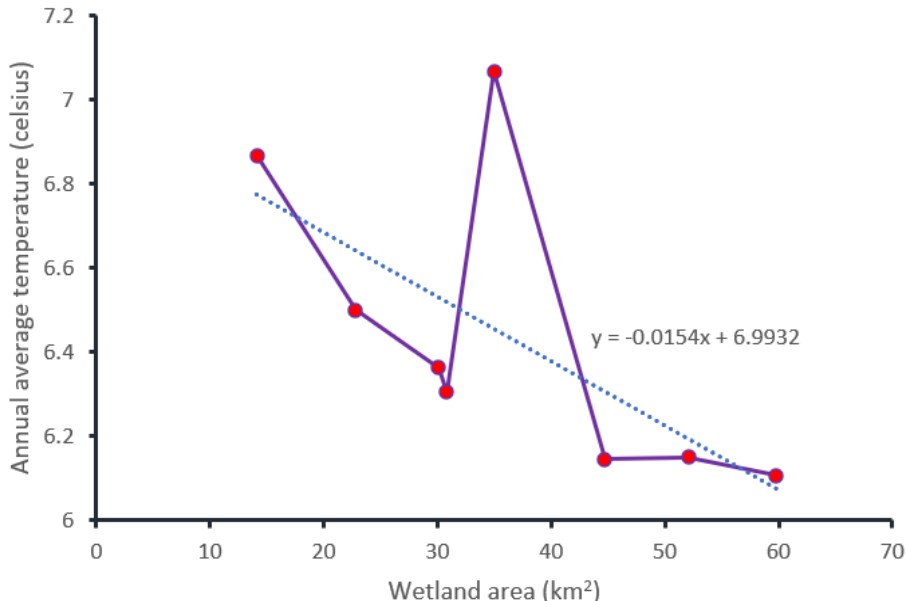

**Figure 10.** The linear relationship between annual average temperature and wetland change.

The relative humidity of Guangling County shows yearly fluctuations (Figure 11). Humidity is the most significant climatic factor, closely tied to wetland changes [46]. The average humidity of the area decreased from the fitting trendline by 4.1% in which a significant amount of wetland resources was lost. To find the linear equation between relative humidity and change of wetland area, using a linear regression method we took the preceding year blocks of relative humidity data corresponding to wetland distribution change in each of eight different years (1984, 1990, 1995, 2000, 2005, 2010, 2015, and 2020) as shown in Figure 12.

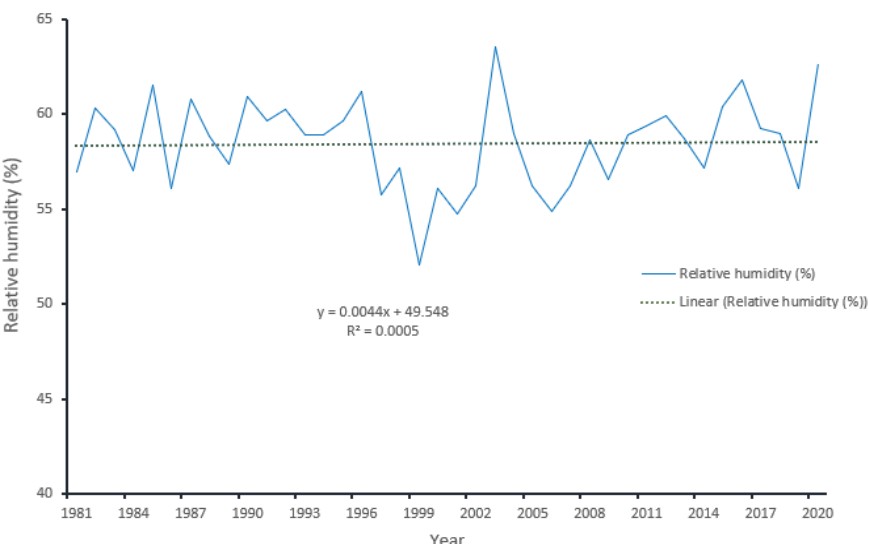

**Figure 11.** Relative humidity of Guangling County, 1981–2020.

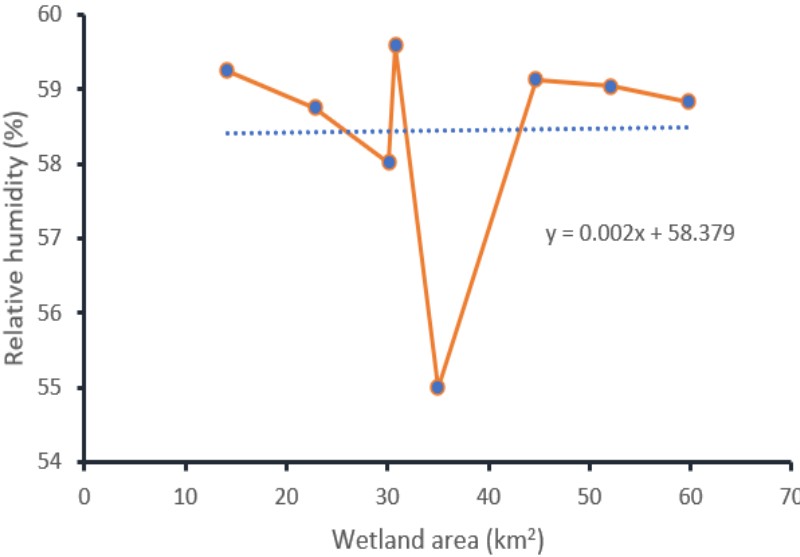

**Figure 12.** The linear relationship between relative humidity and wetland change.

### 3.4. GDP and Wetland Change

The Gross Domestic Product of Guangling County shows a continuous increment (Figure 13). The economy of the county with the value of $R^2$ above 90%. Wetland's change could be directly and indirectly interrelated with the economic growth of one's country. As the number of population growth, the wetlands area substantially stressed in response to economic growth [1]. Building new infrastructures, expansion of agricultural activities, and other human modifications for settlements are the main activities that could degrade the wetland ecosystem in response to economic growth.

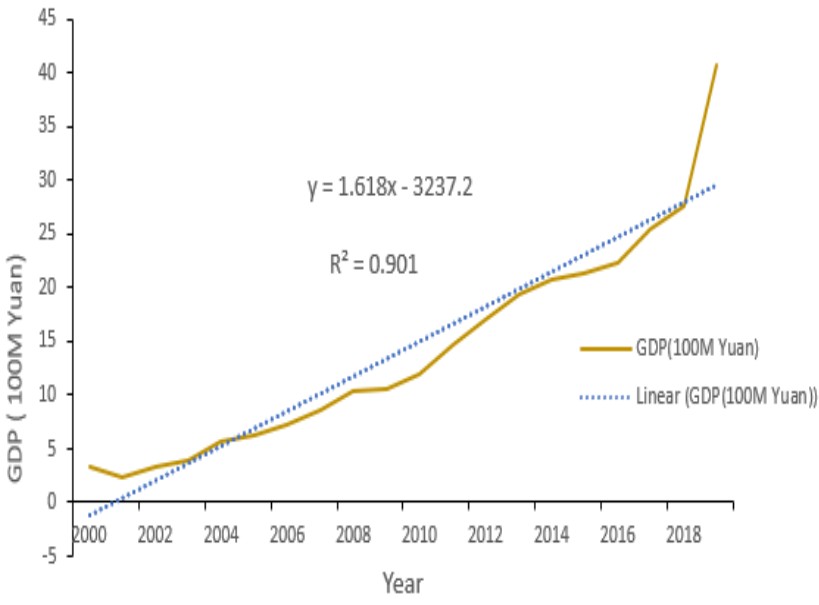

Note that: GDP is Gross Domestic Product

**Figure 13.** GDP of Guangling County.

The annual GDP of the county corresponding to the change of wetland area of that year was taken, Figure 14 shows their linear relationship.

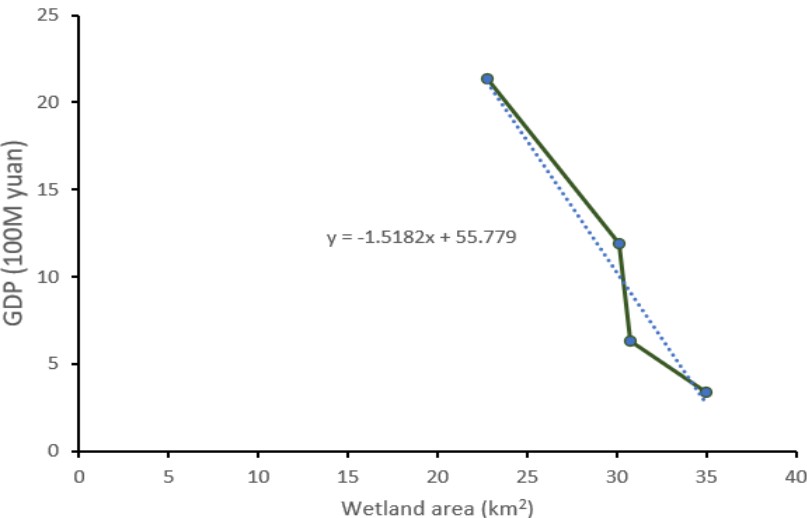

**Figure 14.** The linear relationship of economic data and wetland change.

### 3.5. Grey Correlation Analysis

To calculate the Grey correlation of wetland area in Guangling County and corresponding climatic factors, meteorological data from various sources were used. We selected 4-year blocks of meteorological data—1981, 1982, 1983, and 1984 meteorological data sequence corresponding to wetland distribution data in 1984; 1987, 1988, 1989, and 1990 meteorological data sequence corresponding to wetland distribution data in 1990; 1992, 1993, 1994, and 1995 meteorological data sequence corresponding to wetland distribution data in 1995; 1997, 1998, 1999, and 2000 meteorological sequence data corresponding to wetland distribution data in 2000; 2002, 2003, 2004, and 2005 meteorological data sequence corresponding to wetland distribution data in 2005; 2007, 2008, 2009, and 2010 meteorological data corresponding to wetland distribution data in 2010; 2012, 2013, 2014, and 2015 meteoro-

logical data sequence corresponding to wetland data in 2015; and 2017, 2018, 2019, and 2020 meteorological data sequence corresponding to wetland distribution data in 2020 were used in Grey correlation analysis to calculate the correlation degree of climatic factors and wetland area after normalization and calculation of the mean value, along with relevant meteorological data sequence for the corresponding years of wetland distribution.

The values of grey correlation degree within each of 5-year intervals between climatic variables and wetland area are shown in Table 9. The first line with the values of 0.667, 0.826, and 0.674 reports the degree of grey correlation between annual average temperature (1981–1984) and wetland area (1984), relative humidity (1981–1984) and wetland area (1984), annual rainfall (1981–1984) and wetland area (1984), respectively. The next line with the values of 0.546, 0.874, and 0.647 reports the degree of grey correlation between annual average temperature (1987–1990) and wetland area (1990), relative humidity (1987–1990) and wetland area (1990), annual rainfall (1987–1990) and wetland area (1990), respectively. Similarly, the third line with the values of 0.472, 0.783, and 0.516 reports the degree of grey correlation between annual average temperature (1992–1995) and wetland area (1995), relative humidity (1992–1995), and wetland area (1995), annual rainfall (1992–1995) and wetland area (1995), respectively. The fourth line with the values of 0.740, 0.406, and 0.406 reports the degree of grey correlation between annual average temperature (1997–2000) and wetland area (2000), relative humidity (1997–2000) and wetland area (2000), annual rainfall (1997–2000) and wetland area (2000), respectively. The fifth line with the values of 0.414, 0.590, and 0.515 reports the degree of grey correlation between annual average temperature (2002–2005) and wetland area (2005), relative humidity (2002–2005) and wetland area (2005), annual rainfall (2002–2005) and wetland area (2005), respectively. The sixth line with the values of 0.420, 0.480, and 0.438 reports the degree of grey correlation between annual average temperature (2007–2010) and wetland area (2010), relative humidity (2007–2010) and wetland area (2010), annual rainfall (2007–2010) and wetland area (2010), respectively. The seventh line with the values of 0.420, 0.603, and 0.691 reports the degree of grey correlation between annual average temperature (2012–2015) and wetland area (2015), relative humidity (2012–2015) and wetland area (2015), annual rainfall (2012–2015) and wetland area (2015), respectively. The last line with the values of 0.520, 0.617, and 0.478 reports the degree of grey correlation between annual average temperature (2017–2020) and wetland area (2020), relative humidity (2017–2020) and wetland area (2020), annual rainfall (2017–2020) and wetland area (2020), respectively.

**Table 9.** Grey correlation degree of climatic variables and wetland area within each 5-year interval.

| Climatic Factor | Wetland Area | Climatic Factor | Wetland Area | Climatic Factor | Wetland Area |
|---|---|---|---|---|---|
|  | 0.667 |  | 0.826 |  | 0.674 |
|  | 0.546 |  | 0.874 |  | 0.647 |
| Temperature | 0.472 | Relative humidity | 0.783 |  | 0.516 |
|  | 0.74 |  | 0.406 | Rainfall | 0.406 |
|  | 0.414 |  | 0.589 |  | 0.515 |
|  | 0.42 |  | 0.48 |  | 0.438 |
|  | 0.42 |  | 0.603 |  | 0.691 |
|  | 0.52 |  | 0.617 |  | 0.478 |

Table 10 shows the Grey correlation degree between wetland area and climatic factors calculated using Equation (13).

**Table 10.** The total average correlation degree between climatic variables and wetland area.

| Climatic Factors | Wetland Area |
|---|---|
| Relative Humidity at 2 m | 0.647 |
| Annual Rainfall | 0.546 |
| Annual Average Temperature | 0.525 |

Based on analysis of the grey correlation between wetland area and climatic variables, relative humidity, and wetland area has the highest degree of grey correlation than seen with annual rainfall and annual average temperature, whereas the correlation of annual average temperature has the smallest, indicating that wetland change was positively correlated to relative humidity and negatively correlated to annual average temperature (Table 10). This means that the annual average temperature had a greater negative impact on wetlands resources of the area.

The linkage of a wetland area with humidity, rainfall, temperature, and economic data (GDP) variations can be examined based on their linear relationship. From the linear equations of climatic factors, economic data with wetland area (Figure 8, Figure 10, Figure 12, and Figure 14), we obtain the following equations:

$$
\begin{aligned}
y_h &= 0.002x + 58.38; \\
y_r &= -1.8141x + 461.35; \\
y_t &= -0.0154x + 6.993; \\
y_e &= -1.5182x + 55.78
\end{aligned}
\tag{14}
$$

where $y_h$, $y_r$, $y_t$, and $y_e$ represent humidity, rainfall, temperature, and GDP, respectively. $x$ represents wetland area. Suppose, wetland area ($wa$) = $x$, the preceding equations become:

$$
\begin{aligned}
wa &= 500y_h - 2919; \\
wa &= 254.31 - 0.55y_r; \\
wa &= 454.025 - 64.94y_t; \\
wa &= 36.75 - 0.66y_e
\end{aligned}
\tag{15}
$$

and can be rearranged as:

$$
wa = 500y_h - 64.94y_t - 0.55y_r - 0.66y_e - 28444.89
\tag{16}
$$

Which shows the linkage of wetland change $wa$ with economic growth $y_e$, temperature $y_t$, humidity $y_h$, and rainfall $y_r$.

## 4. Discussion

Several previous studies have demonstrated the potential of classical ML algorithms such as SVM, RF, gradient boosted trees, and CART in wetland mapping and detecting wetland spatial distributions, showing the capacity of machine learning to handle high dimensionality with complex characteristics [50]. Nevertheless, employing a machine learning model for many applications, including classification of satellite imagery and change detection, requires that dataset be established, preferable algorithms selected, computational costs ascertained, user-defined parameter selection, and optimization. Performing machine-learning algorithms in cloud platforms is a modern advance by which Earth observation (EO) can be used for environmental monitoring [17]. Implementing machine learning classifiers on the GEE platform makes this process easier and proved efficient and effective in our experiment.

We employed machine learning models with selected algorithms and parameters. SVM-RBF model displays the most robustness, which outperforms other model performance, producing land cover classification maps and thematic land cover maps whose trends are interpreted in Table 11.

**Table 11.** Wetland (km$^2$) change trends within Guangling County (1984 to 2020).

| Class | 1984 | 1990 | 1995 | 2000 | 2005 | 2010 | 2015 | 2020 | %Change |
|---|---|---|---|---|---|---|---|---|---|
| Wetland | 59.82 | 52.10 | 44.69 | 34.96 | 30.78 | 30.10 | 22.80 | 14.11 | −76.41 |
| Forest | 241.83 | 237.29 | 165.63 | 166.08 | 200.29 | 213.75 | 193.77 | 174.56 | −27.82 |
| Others | 923.23 | 935.49 | 1014.56 | 1023.84 | 993.24 | 981.07 | 1008.31 | 1036.21 | 12.24 |

Note that: % is change in each class over the initial coverage area of that class.

The pixels change detected (Table 8) reveals, 77.99% of the wet areas were converted to other classes and around 3.8% of wet areas were transformed to forests. A total of 45.71 km$^2$ wetland disappeared from 1984 to 2020, and a significant change was seen in the forest (241.83 km$^2$ to 174.56 km$^2$) and other classes (923.23 km$^2$ to 1036.21 km$^2$). Wetlands in the study area continually declined, especially within the first two decades (1984–2005), reaching 30.78 km$^2$. The adjacent 5 years showed no further decrease or increase in wetlands within the area, but within the last decade loss of wetlands reached 14.11 km$^2$, bringing to the degradation percentage of wetlands in the area are around 76.41%. Grey correlation analysis shows the maximum correlation was between wetland area and relative humidity. The minimum was between wetland area and temperature, indicating that wetland ecosystems have a strong connection with humidity and temperature, with decreases in humidity and increases in temperature negatively affecting wetland ecosystems. The maximum increment temperature (by 2.04 °C) was recorded within the interval 1995–2000 when the most wetland resources were lost and the minimum annual average relative humidity was recorded within the interval 1995–2000 when the most wetland resources were lost. From the result, rainfall variations showed no further effects on the loss of wetlands during the specified periods, but a significant decrease in precipitation was seen during the two preceding decades, with the maximum recorded precipitation of 653.1 mm/year, by 89 mm higher than the later.

The resulting linear Equations (15) and (16) showed a strong inverse relation of annual average temperature and economic data with wetland area. Linear relationship and correlation degree of annual average temperature and wetland change, the annual average temperature had the greatest negative impacts on wetlands of the area. The rapid increase in population with economic growth could negatively affect the wetland ecosystem [1,6]. When population increased in response to economic growth, through settlement, new building, and construction of infrastructures and expansion of agriculture, unsustainability, and loss of wetland resources resulted. Temperature shifts and economic growth bringing increased population added to ongoing wetland losses. As a result of human activities in response to economic and population growth; agricultural interventions lead to land reclamation and water diversion from the wetlands, urbanization, and developments of tourism have the main cause for wetland losses.

## 5. Conclusions

In this study, we have shown that GEE has the power to employ an advanced machine learning model and used it to map the long-term dynamicity of wetlands based on time series remote sensing data. SVM performed best at classifying satellite data and detecting changes in time series data. The RF model classifier was highly accurate, but SVM outperformed it because of its higher computational complexity when used for larger and higher-dimensional data sets. The SVM-RBF model outperformed the other four classifiers, allowing successful identification of potential locations of wetlands in the study area and long-term changes to the wetland area.

Understanding the sustainability and status of our natural environment resources is critical, and using them wisely is important. Wetlands are an essential natural terrestrial environment, and their deterioration is a matter of global concern. Previous and present studies show a positive and a negative change to the wetland ecosystem have occurred in China. Restoration of wetlands and wetland ecosystem preservation bring positive change [51]. However, in some parts of the country, an extensive loss of wetland resources was experienced. Guangling County with a coverage area of 1225 km$^2$ has experienced an area of 45.71 km$^2$ wetlands' degradation within the past 3.7 decades. In general, around 81.82% of wetlands area has been lost and 12.33% newly added, showing the wetlands of the study area were found to be undergoing critical change. Climate variations and poor policy implementations, along with economic growth and population increases, have led to a loss of wetlands through settlement, infrastructure construction, and agricultural activities. We conclude that a serious consideration of new policies and implementations

are needed to restore and strengthen the sustainability of wetland resources in Guangling County. Extensive loss of wetlands resulted from less consideration and inadequate policy implementations. The present study can contribute to land cover information and applications for natural resource management and environment studies.

Long-term change detection and monitoring of the earths' features are critical to natural environmental resource management. Although the present study demonstrated effectively classified, land cover thematic maps, and explored change trends associated with climatic change and economy of the study area, their application was limited without detailed land features classifications and delineations. The detailed wetlands monitoring and classifications that can be obtained through high-resolution satellite imagery might allow deep learning to outperform machine learning algorithms. Therefore, the present study could be extended to include detailed monitoring and classifications of wetlands with high spatial resolutions and multispectral satellite imageries using a deep learning model.

**Author Contributions:** Formulation of the study, system, and methods conceptualization, G.F.G.; investigation, X.R.; data collection, methodology, validation, X.R. and G.F.G.; computer code and software execution; G.F.G.; writing—original draft preparation, G.F.G.; writing—review and editing, X.R. and G.F.G.; H.L. contributed a scientific insight in the experiments and analysis. All authors have read and agreed to the published version of the manuscript.

**Funding:** This study was funded by the National Key Research and Development Program of China (Grant No. 2019YFC1804304), the National Natural Science Foundation of China (Grant No. 41771478), the Fundamental Research Funds for the Central Universities (Grant No. 2019B02514).

**Institutional Review Board Statement:** Not applicable.

**Informed Consent Statement:** Not applicable.

**Data Availability Statement:** The long-term historical data Landsat imageries we used are from the U.S. Geological Survey (USGS) and available online at (https://www.usgs.gov/ accessed on 4 April 2021).

**Acknowledgments:** We are grateful to the U.S. Geological survey, they provided the main data set for the study, and all friends and advisors for their valuable help and comments.

**Conflicts of Interest:** There is no conflict of interest from the authors.

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
