# Peer review of "Wetland Change Mapping Using Machine Learning Algorithms, and Their Link with Climate Variation and Economic Growth: A Case Study of Guangling County, China"

_sustainability, doi:10.3390/su14010439_

Round 1
Reviewer 1 Report
The authors aimed to map a wetland in the Guangling County, China and analyzed its changes during the past decades. The paper was within the scope of the journal, while it was poorly written and not well organized. The Figures and Tables were also non-standard. Thus, I suggest the authors substantially modified the manuscript and resubmit it.
Some special comments are as follow,
Tittle: the tittle was wordy, confuse, and difficult for understanding. Please simplify it.
Line 14: in Guangling County - in the Guangling County
Line 15: “This study examines change trends of wetlands based on remotely sensed data 15 while exploring trends associated with climate variations and economic growth in Guangling 16 County, China. ” Please rewrite the sentence.
Line 19: Google Earth Engine GEE - Google Earth Engine (GEE)
Line 24: “over the past 3.7 decades” during which years?
Line 66: “optical ground resolution ”, what does this mean?
Line 76: “cloud-based parallelization https://earthengine.google.com” - cloud-based parallelization (https://earthengine.google.com)
Figure 1: I strongly suggest the authors modify the figure. It was not clear, and administrative boundary was incomplete, for example the China boundary.
I suggest the authors merge section 2.2 and 2.5.
Line 138: “2.3. Data collection and data sources ” - sample data
Line 222: I don’t think introduce the machine learning algorithms in details is necessary here. Please simplify the descriptions on the algorithms, especially reduce the equations.
Line 346: “3.1.1. Classifications using Machine learning classifiers in GEE ” move this section to the method section.
The discussion and conclusion section were also needed to be improved.
Author Response
Dear reviewer of Journal Sustainability, this study was conducted to map wetland change using machine learning algorithms, and to analyze their Link with climate variations and economic growth: A Case Study of Guangling County, China. We are very grateful for your constructive and professional suggestions and comments. We found that suggestions, comments and advice from reviewers improve the quality of our work and the originality of our paper. The authors have adjusted the original considerably and reacted to each of the referees’ comments and questions.
The revision has been done according to the suggestion and advice given by the reviewer.
Point 1: The authors aimed to map a wetland in the Guangling County, China and analysed its changes during the past decades. The paper was within the scope of the journal, while it was poorly written and not well organized. The Figures and Tables were also non-standard. Thus, I suggest the authors substantially modified the manuscript and resubmit it.
Response 1: The manuscript content including language, writing style and organizations, figures, and tables were carefully modified.
Special comments
Point 2: Tittle: the tittle was wordy, confuse, and difficult for understanding. Please simplify it.
Response 2: The tittle of the paper was simplified, to reduce confusion.
Point 3: Line 14: in Guangling County - in the Guangling County
Response 3: Corrected
Point 4: Line 15: “This study examines change trends of wetlands based on remotely sensed data while exploring trends associated with climate variations and economic growth in Guangling County, China.” Please rewrite the sentence.
Response 4: This study examines change of wetlands area based on remotely sensed data while exploring trends associated with climate variations and economic growth in Guangling County, China.
Point 5: Line 19: Google Earth Engine GEE - Google Earth Engine (GEE)
Response 5: Corrected
Point 6: Line 66: “optical ground resolution”, what does this mean?
Response 6: It has been corrected to “optimal ground resolution”
Point 7: Line 76: “cloud-based parallelization https://earthengine.google.com” - cloud-based parallelization (https://earthengine.google.com)
Response 7: Corrected
Point 8: Figure 1: I strongly suggest the authors modify the figure. It was not clear, and administrative boundary was incomplete, for example the China boundary.
Response 8: Modified
Point 9: I suggest the authors merge section 2.2 and 2.5.
Response 9: Merged in one section
Point 10: Line 138: “2.3. Data collection and data sources ” - sample data
Response 10: Corrected
Point 11: Line 222: I don’t think introduce the machine learning algorithms in details is necessary here. Please simplify the descriptions on the algorithms, especially reduce the equations.
Response 11: The descriptions and the equations have been reduced and briefly introduced.
Point 12: Line 346: “3.1.1. Classifications using Machine learning classifiers in GEE” move this section to the method section.
Response 12: Introducing machine learning classifiers and their brief theory has been included in methods section. The classification results from the models were presented under result section.
Point 13: The discussion and conclusion section were also needed to be improved.
Response 13: The Discussion and conclusion have improved and concised.
Reviewer 2 Report
Generally, this investigation report is presented in accurate way. All components of the structure of this paper (motivation (Introduction), Materials and Methods, Results, Discussion and Conclusions) are clear and concrete. Methodological combination for the analysis of time series assure the confidence of the results.
Some questions and suggestions are as follows:
- The content of the article is complex and not concise, which is not conducive to reading and understanding. There are a lot of graphs and tables in the article, which can be deleted or synthesized appropriately.
- The “Introduction” of the article should be adjusted to highlight the lack of wetland research in areas without data or on county scale. The fifth paragraph in “Introduction” should not only focus on Guangling County, but to those areas with very limited wetland information, otherwise your research is too local.
- The chapter on research methods is too lengthy. Such as, the section 2.2, 2.3 and 2.4 are redundant and can be combined and briefly introduced.
- The section 3.3.4 (GDP and wetland change) should not be included in the section 3.3 (Climate variation and wetland change).
- The statistical chart in the article is very ugly and needs to be carefully revised.
- The section 3.4 (Grey correlational analysis) should be introduced in the Methods。
- The discussion on the relationship between sustainability of wetlands and economy and population can be appropriately extended.
- The conclusion of the article is neither concise nor focused, and needs to be carefully revised.
- There are many grammatical and lexical errors in the article.
Author Response
Dear reviewer of Journal Sustainability, this study was conducted to map wetland change using machine learning algorithms, and to analyze their Link with climate variations and economic growth: A Case Study of Guangling County, China. We are very grateful for your constructive and professional suggestions and comments. We found that suggestions, comments and advice from reviewers improves the quality of our work and the originality of our paper. The authors have adjusted the original considerably and reacted to each of the referees’ comments and questions.
The revision has been done according to the suggestion and advice given by the reviewer.
Generally, this investigation report is presented in accurate way. All components of the structure of this paper (motivation (Introduction), Materials and Methods, Results, Discussion and Conclusions) are clear and concrete. Methodological combination for the analysis of time series assures the confidence of the results.
Questions and suggestions
Point 1: The content of the article is complex and not concise, which is not conducive to reading and understanding. There are a lot of graphs and tables in the article, which can be deleted or synthesized appropriately.
Response 1: The article has been carefully revised to reduce the complexity, making it simpler for the readers. Figures and tables have been reduced and concisely presented.
Point 2: The “Introduction” of the article should be adjusted to highlight the lack of wetland research in areas without data or on county scale. The fifth paragraph in “Introduction” should not only focus on Guangling County, but to those areas with very limited wetland information, otherwise your research is too local.
Response 2: The articles highlighted and described the previous research gap and lack of information in the Guangling County and in the “introduction section” of 5th paragraph wetland information limitation, both regionally and internationally introduced.
Point 3: The chapter on research methods is too lengthy. Such as, the section 2.2, 2.3 and 2.4 are redundant and can be combined and briefly introduced.
Response 3: They have been merged in one section and briefly introduced.
Point 4: The section 3.3.4 (GDP and wetland change) should not be included in the section 3.3 (Climate variation and wetland change).
Response 4: Corrected! “Gross Domestic Product (GDP) and wetland change” has been separately presented.
Point 5: The statistical chart in the article is very ugly and needs to be carefully revised.
Response 5: All figures and tables have been neatly and concisely presented in the new version of the article.
Point 6: The section 3.4 (Grey correlational analysis) should be introduced in the Methods
Response 6: Grey correlational analysis introduced in the methods section.
Point 7: The discussion on the relationship between sustainability of wetlands and economy and population can be appropriately extended.
Response 7: The linkage or relationship of economy and population with the sustainability of wetlands were presented.
Point 8: The conclusion of the article is neither concise nor focused, and needs to be carefully revised
Response 8: The conclusion of the article has been briefly summarized.
Point 9: There are many grammatical and lexical errors in the article
Response 9: The spelling and grammatical error was carefully edited and corrected.
Reviewer 3 Report
Manuscript ID: sustainability-1491278
Title: Mapping Wetland Change Trends Using Machine Learning Algorithms, and Their Link With Climate Variation and Economic Growth: A Case Study of Guangling County, China.
OVERVIEW
The manuscript examines the change of wetlands area based on remotely sensed data while exploring trends associated with climate variations and economic growth in Guangling County, China and explores how factors such as county economic growth (GDP), humidity, and temperature variations are tightly linked with wetland change.
The subject matter is actual, interesting and within the scope of the Journal Sustainability.
The title fully describes the manuscript.
The English is good.
As for the rest, I have some revisions to suggest.
In conclusion, I believe this manuscript is worthy of publication, after a major revision.
SPECIFIC COMMENTS
Line 110: Where reads “The study area has is a continental monsoon climate”, should read “The study area has a continental monsoon climate”.
Line 242, Eq. 5, 6 and 7: the authors must explain the meaning of all variables in the equations.
Line 268: The link https://www.ece.uvic.ca/~aalbu/computer%20vision%202009/gonzalez%20-%20mini- 268 mum%20distance%20classifier.pdf does not work.
Line 290: The text after the equation should be in another text line.
Line 440: The average variation of annual precipitation is rather small. Please try to correlate the wetland area with the change in the rainfall distribution over the year.
Line 481: Please renumber the equation.
Line 503: Please consider the new equation number.
Line 504, Table 10: The table is ambiguous to read. Please define the meaning for the values in each line. Reports to a 5-year interval?
Line 507: Where reads “indicating a stronger connection between humidity variation and wetland change”. Please explain. Figure 11 reports a very small variation in the average relative humidity over the decades. According to the authors, it is the cause or the consequence of the change in the wetland area? It is not very clear in the text.
Lines 511 to 520: It is an oversimplification to represent such equations. For example, economic growth implies more agricultural area for crop production or is related to tourism or other services. In my opinion, the authors should disaggregate the economic growth in different sectors of activity and try to obtain a correlation with the economic growth of activities that have a direct impact on the wetlands.
Line 518, Eq I: Please renumber the equations in ascending order from the beginning of the manuscript.
Line 518, Eq I: The humidity (h) has a factor of 500 in the equation, however from Figure 10, the relative humidity does not have a strong variation in the region. Does it imply that a very small variation in the average relative humidity (1% to 2%) is a consequence or the cause of the decrease in the wetland area? Irrigation also contributes to soil moisture, evapotranspiration and atmospheric relative humidity. Please improve the justification.
Line 536, table 12: Please align the columns of the table.
Line 536, table 12: Please explain the %Change calculation. (14.11 - 59.82)/59.82 * 100 = -76.41% or -2.18%/year. In table is -3.73%!!
Line 545 reads: “The extensive loss of wetland resulted from temperature variations in the region and indirectly from the economic growth of the county.” Climate change and temperature increase have a global trend and local variabilities. Please improve the justification.
Author Response
Dear reviewer of Journal Sustainability, this study was conducted to map wetland change using machine learning algorithms, and to analyze their Link with climate variations and economic growth: A Case Study of Guangling County, China. We are very grateful for your constructive and professional suggestions and comments. We found that suggestions, comments, and advice from reviewers improves the quality of our work and the originality of our paper. The authors have adjusted the original considerably and reacted to each of the referees’ comments and questions.
The revision has been done according to the suggestion and advice given by the reviewer.
Manuscript ID: sustainability-1491278
Title: Mapping Wetland Change Trends Using Machine Learning Algorithms, and Their Link with Climate Variation and Economic Growth: A Case Study of Guangling County, China.
OVERVIEW
The manuscript examines the change of wetlands area based on remotely sensed data while exploring trends associated with climate variations and economic growth in Guangling County, China and explores how factors such as county economic growth (GDP), humidity, and temperature variations are tightly linked with wetland change.
The subject matter is actual, interesting and within the scope of the Journal Sustainability.
The title fully describes the manuscript.
The English is good.
As for the rest, I have some revisions to suggest.
In conclusion, I believe this manuscript is worthy of publication, after a major revision.
SPECIFIC COMMENTS
Point 1: Line 110: Where reads “The study area has is a continental monsoon climate”, should read “The study area has a continental monsoon climate”.
Response 1: Corrected to “The study area has a continental monsoon climate”.
Point 2: Line 242, Eq. 5, 6 and 7: the authors must explain the meaning of all variables in the equations.
Response 2: The meaning of each variables in each equation were explained.
Point 3: Line 268: The link https://www.ece.uvic.ca/~aalbu/computer%20vision%202009/gonzalez%20-%20mini- 268 mum%20distance%20classifier.pdf does not work.
Response 3: Similar suggestions came from other experts. Since the method can be introduced without it, the link has been deleted for the sake of simplifications and the method was introduced with the help of other source.
Point 4: Line 290: The text after the equation should be in another text line.
Response 4: The text after the equation has been written in another text line.
Point 5: Line 440: The average variation of annual precipitation is rather small. Please try to correlate the wetland area with the change in the rainfall distribution over the year.
Response 5: The preceding years of annual rainfall data sequence corresponding to wetland distribution data were correlated. For instance, 3 years preceding blocks of rainfall data—1981, 1982, 1983 including 1984 has the long-term effect on wetland distributions seen in 1984, thus, the annual average rainfall data sequence corresponding to wetland distribution data in 1984 was correlated.
Point 6: Line 481: Please renumber the equation.
Response 6: The equation has renumbered.
Point 7: Line 503: Please consider the new equation number.
Response 7: The new equation number was considered.
Point 8: Line 504, Table 10: The table is ambiguous to read. Please define the meaning for the values in each line. Reports to a 5-year interval?
Response 8: The values in each line reports the correlation degree between climatic variabilities and wetland area in each 5-year interval. For instance, the first line with the values of 0.667, 0.8261, and 0.674 shows the degree of grey correlation between annual average temperature and wetland area, relative humidity and wetland area, annual rainfall, and wetland area respectively.
Point 9: Line 507: Where reads “indicating a stronger connection between humidity variation and wetland change”. Please explain. Figure 11 reports a very small variation in the average relative humidity over the decades. According to the authors, it is the cause or the consequence of the change in the wetland area? It is not very clear in the text.
Response 9: From the analysis of grey correlation, humidity shows the greatest degree indicating the positive connection between with wetland area. Annual average humidity of Guangling County reports instabilities in each year of the study period, that has direct connection with wetland change. For example; from our experiment, the greater variability of annual average humidity was recorded within the interval of 1995–2000, when the most wetland resources were lost.
Note that: the relative humidity has presented in %, in 1995 measured was 59.31% which is dropped to 51.44% in 1999, the fluctuation was up to 7.87% when the maximum wetland change was detected.
Point 10: Lines 511 to 520: It is an oversimplification to represent such equations. For example, economic growth implies more agricultural area for crop production or is related to tourism or other services. In my opinion, the authors should disaggregate the economic growth in different sectors of activity and try to obtain a correlation with the economic growth of activities that have a direct impact on the wetlands.
Response 10: As a result of human activities in response to economy and population growth; agricultural interventions lead to land reclamation and water diversion from the wetlands, urbanization and developments of tourism have the main cause for wetland losses. Their linear relationship reveals the economy of the county had negative impacts on wetland area.
Point 11: Line 518, Eq I: Please renumber the equations in ascending order from the beginning of the manuscript.
Response 11: All the equations in article have been renumbered in ascending order.
Point 12: Line 518, Eq I: The humidity (h) has a factor of 500 in the equation, however from Figure 10, the relative humidity does not have a strong variation in the region. Does it imply that a very small variation in the average relative humidity (1% to 2%) is a consequence or the cause of the decrease in the wetland area? Irrigation also contributes to soil moisture, evapotranspiration and atmospheric relative humidity. Please improve the justification.
Response 12: There is a variation in relative humidity over the year, as a result, relative humidity revealed highest correlation and linear relationship with wetland area. From our analysis the minimum humidity measured in each half decade the maximum wetland lost, indicating that wetland ecosystems have a strong connection with humidity.
Point 13: Line 536, table 12: Please align the columns of the table.
Response 13: Corrected!
Point 14: Line 536, table 12: Please explain the %Change calculation. (14.11 - 59.82)/59.82 * 100 = -76.41% or -2.18%/year. In table is -3.73%!!
Response 14: The percentage change was calculated considering the “change in each class over the entire of the county”. Now it has been corrected to” change in each class over the initial coverage area of that class”.
Point 15: Line 545 reads: “The extensive loss of wetland resulted from temperature variations in the region and indirectly from the economic growth of the county.” Climate change and temperature increase have a global trend and local variabilities. Please improve the justification.
Response 15: The economy of the county and annual average temperature were negatively related to wetland area. The annual average temperature showed the least correlated with wetland, indicating that increases in temperature had negatively impacted wetlands. The maximum increment temperature (by 2.04 °C) was recorded within the interval 1995–2000 when the most wetland resources were lost.
Round 2
Reviewer 1 Report
The manuscript was improved by authors. Several small suggestions are as follows,
Abstract
This study examines change of wetlands area based on remotely sensed data while exploring trends associated with climate variations and economic growth in Guangling County, China. - This study examines change of wetlands area based on remotely sensed data, and explores the trend of wetlands associated with climate variations and economic growth in Guangling County, China.
Figure 2 is not clear. Can you replace it with a high resolution. In addition, the caption is incomplete. General workflow. - The workflow of the proposed wetland mapping.
Several recently published closely related papers should be cited. Such as,
Wang et al., 2020. Mapping coastal wetlands of China using time series Landsat images in 2018 and Google Earth Engine.
Sun et al., 2020. Mapping Coastal Wetlands of the Bohai Rim at a Spatial Resolution of 10 m Using Multiple Open-Access Satellite Data and Terrain Indices.
Author Response
Dear reviewer of Journal Sustainability, this study was conducted to map wetland change using machine learning algorithms, and to analyze their Link with climate variations and economic growth: A Case Study of Guangling County, China. Thank you for the additional suggestions and comments. The authors have adjusted the original considerably and reacted to each of the referees’ comments and questions.
The revision has been done according to the suggestion and advice given by the reviewer.
The manuscript was improved by authors. Several small suggestions are as follows,
Abstract
This study examines change of wetlands area based on remotely sensed data while exploring trends associated with climate variations and economic growth in Guangling County, China. - This study examines change of wetlands area based on remotely sensed data, and explores the trend of wetlands associated with climate variations and economic growth in Guangling County, China.
Point 1: Figure 2 is not clear. Can you replace it with a high resolution. In addition, the caption is incomplete. General workflow. - The workflow of the proposed wetland mapping.
Response 1: Figure 2- has been modified and replaced by a high-resolution and the caption has been edited.
Point 2: Several recently published closely related papers should be cited. Such as,
Wang et al., 2020. Mapping coastal wetlands of China using time series Landsat images in 2018 and Google Earth Engine.
Sun et al., 2020. Mapping Coastal Wetlands of the Bohai Rim at a Spatial Resolution of 10 m Using Multiple Open-Access Satellite Data and Terrain Indices.
Response 2: Since the suggested published papers are methodically related to our paper we have referred to and cited them.
Reviewer 2 Report
Generally speaking, this article has many deficiencies in both form and content. At present, it has not reached the goal of revision. I still have some questions and suggestions for the authors:
(1) The Y-axis and X-axis of all figures have no scale marks.
(2) There is no description of the results below Chapter 3.4 (GDP and wetland change). I don't know what's going on. At this point, I can't convince myself that this article has been carefully revised.
(3) There is no need to split Chapter 3.3 into three small sections, which makes the article look very messy. I strongly recommend merging Section 3.3.1, Section 3.3.2 and Section 3.3.3. On this basis, the figures 6, 8 and 10 can be incorporated into one figure, and also figures 7, 9 and 11 can be combined into one figure.
(4) Why do the numbers in Table 11 retain so many decimal places?
(5) I've never seen multiple equations listed in one line. (line 553 and 558)
The charts of the article are too complicated to meet the standards of professional journals. I think the article should be carefully revised after the review. All authors, especially corresponding authors, should make the final check on the article. Obviously, this article is not doing well in this regard.
Author Response
Dear reviewer of Journal Sustainability, this study was conducted to map wetland change trends using machine learning algorithms, and to analyze their Link with climate variation and economic growth: A Case Study of Guangling County, China. We are very grateful for your constructive and professional suggestions and comments. We found that suggestions, comments, and advice from reviewers improves the quality of our work and the originality of our paper. The authors have adjusted the original considerably and reacted to each of the referees’ comments and questions.
The revision has been done according to the suggestion and advice given by the reviewer.
Generally speaking, this article has many deficiencies in both form and content. At present, it has not reached the goal of revision. I still have some questions and suggestions for the authors:
Point 1: (1) The Y-axis and X-axis of all figures have no scale marks.
Response 1: All figures have been modified with scale marks.
Point 2: (2) There is no description of the results below Chapter 3.4 (GDP and wetland change). I don't know what's going on. At this point, I can't convince myself that this article has been carefully revised.
Response 2: Previously, we described how Economic data and wetland change were correlated under the discussion part. Now also been given under the sub-topic (GDP and wetland change).
Point 3: (3) There is no need to split Chapter 3.3 into three small sections, which makes the article look very messy. I strongly recommend merging Section 3.3.1, Section 3.3.2 and Section 3.3.3. On this basis, the figures 6, 8 and 10 can be incorporated into one figure, and also figures 7, 9 and 11 can be combined into one figure.
Response 3: Merged! Sections 3.3.1, 3.3.2, and 3.3.3 were merged into one section. As a result, figures 6, 8, and 10 have been combined by one figure, and figures 7, 9, and 11 have also been combined by one figure.
Point 4: (4) Why do the numbers in Table 11 retain so many decimal places?
Response 4: Edited to a few decimal places.
Point 5: (5) I've never seen multiple equations listed in one line. (line 553 and 558)
Response 5: The equations were driven from the linear regression method, showing the linear relationship between wetland change and climatic variabilities (annual average temperature, annual rainfall, and relative humidity). The equations have been now written in separate lines for easier readability.
Point 6: The charts of the article are too complicated to meet the standards of professional journals. I think the article should be carefully revised after the review. All authors, especially corresponding authors, should make the final check on the article. Obviously, this article is not doing well in this regard.
Response 6: The tables and figures in the paper have been edited, for each table, charts, and figures the interpretation and definition have been given in the text, which could be simpler for the readers. The article has been revised by all authors.
Reviewer 3 Report
Manuscript ID: sustainability-1491278
Title: Wetland Change Mapping Using Machine Learning Algorithms, and Their Link with Climate Variation and Economic Growth: A Case Study of Guangling County, China.
The authors addressed all the questions made in the first review and improved the manuscript. In my opinion, this manuscript is worthy of publication, as it is.
Author Response
Dear reviewer of Journal Sustainability, this study was conducted to map wetland change using machine learning algorithms, and to analyze their Link with climate variations and economic growth: A Case Study of Guangling County, China.
We thank you for your professional comments. The article has been revised!
Round 3
Reviewer 2 Report
The introduction of the sources of climatic variables in section 3.3 should be moved to the former section “2. Materials and Methods”.
In the section 3.5, all numbers should be uniformly reserved with 3 significant digits after the decimal point, not only in the table, but also in the text.
For figure 6 and 7, because the units of climatic variables such as rainfall, temperature and humidity are different, their inter-annual variations should not be drawn in the same coordinate system. What I meant in the last round of comments was suggesting you combine the annual variations of the three variables into a graph in the form of three sub-graphs.
It is necessary to introduce Figure 6 before Figure 7 in the text. (lines 463-465). In addition, the result description of Climate variation and wetland change is not sufficient. The same is true for in the result description of GDP and wetland change.
The application of characters in formulas (lines 549-557) is not in a normal way. The dependent variables of humidity, rainfall, temperature, and GDP should use different symbols such as yh, yr, yt, and yG. Each formula should be listed in a separate line.
In fact, all these comments are mainly about the basic writing norms, but it is obvious that the authors have not done a good job in this.
Author Response
Dear reviewer of Journal Sustainability, this study was conducted to map wetland change using machine learning algorithms, and to analyze their Link with climate variations and economic growth: A Case Study of Guangling County, China. Thank you for the additional suggestions and comments. The authors have considered the suggestions and comments.
Point 1: The introduction of the sources of climatic variables in section 3.3 should be moved to the former section “2. Materials and Methods”.
Response 1: The source of climatic time-series data that should be acknowledged was moved to section 2.2
Point 2: In the section 3.5, all numbers should be uniformly reserved with 3 significant digits after the decimal point, not only in the table, but also in the text.
Response 2: Corrected!
Point 3: For figure 6 and 7, because the units of climatic variables such as rainfall, temperature and humidity are different, their inter-annual variations should not be drawn in the same coordinate system. What I meant in the last round of comments was suggesting you combine the annual variations of the three variables into a graph in the form of three sub-graphs.
Response 3: The three climatic variables have different units of measure. The figures (6&7) were plotted aimed to show correlations between climate factors in correspondence to wetland distribution change within the selected periods only (1984-2020), disregarding the units of measure. The original time-series trends have been shown within different coordinates.
Point 4: It is necessary to introduce Figure 6 before Figure 7 in the text. (lines 463-465). In addition, the result description of Climate variation and wetland change is not sufficient. The same is true for in the result description of GDP and wetland change.
Response 4: Additional description given.
Point 5: The application of characters in formulas (lines 549-557) is not in a normal way. The dependent variables of humidity, rainfall, temperature, and GDP should use different symbols such as yh, yr, yt, and yG. Each formula should be listed in a separate line.
Response 5: Normalized and written in separate text lines.